# Can we decode phonetic features in inner speech using surface electromyography?

**Ladislas Nalborczyk**[1,2]*, **Romain Grandchamp**[1], **Ernst H. W. Koster**[2], **Marcela Perrone-Bertolotti**[1,3], **Hélène Lœvenbruck**[1]

**1** Univ. Grenoble Alpes, CNRS, LPNC, Grenoble, France, **2** Department of Experimental Clinical and Health Psychology, Ghent University, Ghent, Belgium, **3** Institut Universitaire de France (IUF), Paris, France

* ladislas.nalborczyk@univ-grenoble-alpes.fr

**Data Availability Statement:** Pre-registered protocol, preprint, data, as well as reproducible code and figures are available at: https://osf.io/czer4/.

## Abstract

Although having a long history of scrutiny in experimental psychology, it is still controversial whether wilful inner speech (covert speech) production is accompanied by specific activity in speech muscles. We present the results of a preregistered experiment looking at the electromyographic correlates of both overt speech and inner speech production of two phonetic classes of nonwords. An automatic classification approach was undertaken to discriminate between two articulatory features contained in nonwords uttered in both overt and covert speech. Although this approach led to reasonable accuracy rates during overt speech production, it failed to discriminate inner speech phonetic content based on surface electromyography signals. However, exploratory analyses conducted at the individual level revealed that it seemed possible to distinguish between rounded and spread nonwords covertly produced, in two participants. We discuss these results in relation to the existing literature and suggest alternative ways of testing the engagement of the speech motor system during wilful inner speech production.

## Introduction

As you read these words, you may be experiencing the presence of a familiar speechlike companion. This internal speech production may accompany daily activities such as reading (see [1–4], but see [5, 6]), writing ([7]), memorising ([8, 9]), future planning [8], problem solving [9, 10] or musing (for reviews see [11–14]). Several studies using experience sampling or questionnaires (e.g., [15, 16]) have shown that by deliberately paying attention to this internal speech, one can examine its phenomenological properties such as identity (whose voice is it?) or other high-level characteristics (e.g., is it gendered?). Moreover, it is often possible to examine lower-level features like the tone of the inner speech, its pitch or its tempo. This set of basic observations leads to some important insights about the nature of inner speech. The simple fact that we can make sensory judgements about our inner speech tautologically reveals that inner speech is accompanied by sensory percepts (e.g., speech sounds, kinaesthetic feelings). Some of these introspective accounts have been examined, tested and complemented using empirical methods from cognitive neuroscience. As summarised in [17], behavioural and

**Funding:** This project was funded by the ANR project INNERSPEECH (grant number ANR-13-BSH2-0003-01). The first author of this manuscript was funded by a fellowship from Univ. Grenoble Alpes.

**Competing interests:** The authors have declared that no competing interests exist.

neuroimaging data reveal that some variants of inner speech are associated with auditory and/ or somatosensory sensations that are reflected by auditory and/or somatosensory cortex activations. Visual representations may also be at play, typically for inner language in the deaf population. Inner verbalising therefore involves the reception of imaginary multisensory signals. This leads to other fundamental questions: where do these percepts come from? Why do they sound and feel like the ones we experience when we actually (overtly) speak?

Two main classes of explanatory theories have been offered to answer these questions. A first class of theories, that derives from Vygotky's views on language and thought, and that we describe as the *abstraction view* [18], suggest that inner speech is profoundly internalised, abbreviated and condensed in form. Vygotsky suggested that inner speech evolved from so-called egocentric speech (i.e., self-addressed overt speech or private speech), via a gradual process of internalisation during childhood [19]. According to him, the properties of speech are transformed during this internalisation, and inner speech cannot merely be described as a weakened form of overt speech (as claimed for instance by [20]). This has led some scholars to conceive of inner speech as predominantly pertaining to semantics, excluding any phonological, phonetic, articulatory, or even auditory properties (e.g., [18, 21, 22]). The property of abbreviation and condensation is supported by several psycholinguistic experiments on the qualitative and quantitative differences between overt and covert speech, as concerns rate and error biases (e.g., [18, 22–24], but see [25]). Such condensation implies that the auditory qualities mentioned above would only rarely be observed during introspection and would merely be the result of learned associations between abstract linguistic representations and auditory percepts. A second class of theories is described under the umbrella term of *motor simulation view*. These theories suggest that inner speech can be conceived as a kind of action on its own [26, 27], produced in the same way as overt speech is, except that the last stage of articulatory execution is only simulated. Most theories under this view share the postulate that the speech motor system is involved (to some extent) during inner speech production and that the auditory and somatosensory consequences of the simulated articulatory movements constitute the inner speech percepts referred to in subjective studies.

As explained in the ConDialInt model [28], these two views can be reconciled if various degrees of unfolding of inner speech are considered. Fully condensed forms of inner speech only involve semantics, and are deprived of the acoustic, phonological and syntactic qualities of overt speech. Expanded forms of inner speech, on the other hand, presumably engage prosodic and morpho-syntactic formulation as well as phonological specification, articulatory simulation and the perception of an inner voice. Between the fully condensed abstract forms and the expanded articulation-ready form, it can be assumed that various semi-condensed forms may exist, with morphosyntactic properties and perhaps even phonological features, depending on the stage at which the speech production process is truncated. Such a view was also taken by [29] who has suggested that inner speech varies with cognitive demands and emotional conditions on a continuum between extremely condensed and expanded forms (see also [11, 27]). Therefore, the two views of inner speech (abstraction vs. simulation) can be construed as descriptions of two opposite poles on the condensation dimension. On the most expanded side of the continuum, inner speech entails full phonetic specification and articulatory simulation. It might therefore be expected that speech motor activity could be detectable. If the motor simulation view is correct, then motor activity could be recorded during expanded forms of inner speech. If, on the other hand, the abstraction view applies to all forms of inner speech, then no motor activity should be present, even in phonologically-expanded forms.

Previous research has demonstrated that it is possible to record muscle-specific electromyographic correlates of inner speech (e.g., [30, 31]). However, these studies mostly focused on

small samples of participants and sometimes used invasive intramuscular electromyography. In contrast, more recent research studies using surface electromyography lead to mixed results (e.g., [32]). Building upon previous work, we describe an experimental set-up using surface electromyography with the aim of testing the involvement of specific speech muscle groups during the covert production of phonologically expanded speech forms.

## Inner speech as motor imagery of speech

Speech production is a complex motor action, involving the fine-grained coordination of more than 100 muscles in the upper part of the body [33]. In adult humans, its covert counterpart (referred to as *inner speech* or *verbal imagery*) has developed to support a myriad of different functions. In the same way as visual imagery permits to mentally examine visual scenes, *verbal imagery* can be used as an internal tool, allowing –amongst other things– to rehearse or to prepare past or future conversations [11, 14]. Because speech production results from sequences of motor commands which are assembled to reach a given goal, it belongs to the broader category of motor actions [34]. Therefore, a parallel can be drawn between verbal imagery and other forms of motor imagery (e.g., imagined walking or imagined writing). Accordingly, studies on the nature of inner speech might benefit from insights gained from the study of motor imagery and the field of motor cognition [34, 35].

Motor imagery can be defined as the mental process by which one rehearses a given action, without engaging in the physical movements involved in this particular action. One of the most influential theoretical accounts of this phenomenon is the *motor simulation theory* [34, 36, 37]. In this framework, the concept of simulation refers to the "offline rehearsal of neural networks involved in specific operations such as perceiving or acting" [34]. The MST shares some similarities with the theories of embodied and grounded cognition [38] in that both account for motor imagery by appealing to a simulation mechanism. However, the concept of simulation in grounded theories is assumed to operate in order to acquire specific conceptual knowledge [39], which is not the concern of the MST. In other words, we should make a distinction between *embodiment of content*, which concerns the semantic content of language, and *embodiment of form*, which concerns "the vehicle of thought", that is, proper verbal production [40].

A second class of explanatory models of motor imagery are concerned with the phenomenon of *emulation* and with *internal models* [41]. Internal model theories share the postulate that action control uses internal models, that is, systems that simulate the behaviour of the motor apparatus [42, 43]. The function of internal models is to estimate and anticipate the outcome of a motor command. Among the internal model theories, motor control models based on robotic principles [44, 45] assume two kinds of internal models (that are supposed to be coupled and regulated): a forward model (or simulator) that predicts the sensory consequences of motor commands from efference copies of the issued motor commands, and an inverse model (or controller) that calculates the feedforward motor commands from the desired sensory states [17, 41].

Emulation theories [46, 47] borrow from both simulation theories and internal model theories and provide operational details of the simulation mechanism. In the emulation model proposed by [46], the *emulator* is a device that implements the same input-output function as the body (i.e., the musculoskeletal system and relevant sensory systems). When the emulator receives a copy of the control signal (which is also sent to the body), it produces an output signal (the emulator feedback), identical or similar to the feedback signal produced by the body, yielding mock sensory percepts (e.g., visual, auditory, kinaesthetic) during motor imagery.

By building upon models of speech motor control [45, 48], a recent model describes wilful (voluntary) expanded inner speech as "multimodal acts with multisensory percepts stemming from coarse multisensory goals" [17]. In other words, in this model the auditory and kinaesthetic sensations perceived during inner speech are assumed to be the predicted sensory consequences of simulated speech motor acts, emulated by internal forward models that use the efference copies of motor commands issued from an inverse model [46]. In this framework, the peripheral muscular activity recorded during inner speech production is assumed to be the result of *partially* inhibited motor commands. It should be noted that both simulation, emulation, and motor control frameworks can be grouped under the *motor simulation view* and altogether predict that the motor system should be involved to some extent during motor imagery, and by extension, during inner speech production. We now turn to a discussion of findings related to peripheral muscular activity during motor imagery and inner speech.

## Electromyographic correlates of covert actions

Across both simulationist and emulationist frameworks, motor imagery has consistently been defined as the mental rehearsal of a motor action without any overt movement. One consequence of this claim is that, in order to prevent execution, the neural commands for muscular contractions should be blocked at some level of the motor system by active inhibitory mechanisms [49]. Despite these inhibitory mechanisms, there is abundant evidence for peripheral muscular activation during motor imagery [49–51]. As suggested by [36], the incomplete inhibition of the motor commands would provide a valid explanation to account for the peripheral muscular activity observed during motor imagery. This idea has been corroborated by studies of changes in the excitability of the motor pathways during motor imagery tasks [52]. For instance, [53] measured spinal reflexes while participants were instructed to either press a pedal with the foot or to simulate the same action mentally. They observed that both H-reflexes and T-reflexes increased during motor imagery, and that these increases correlated with the force of the simulated pressure. Moreover, the pattern of results observed during motor imagery was similar (albeit weaker in amplitude) to that observed during execution, supporting the *motor simulation view* of motor imagery. Using transcranial magnetic stimulation, several investigators observed muscle-specific increases of motor evoked potentials during various motor imagery tasks, whereas no such increase could be observed in antagonist muscles [54, 55].

When considered as a form of motor imagery, inner speech production is also expected to be accompanied with peripheral muscular activity in the speech muscles. This idea is supported by many studies showing peripheral muscular activation during inner speech production [10, 30, 31, 56–58], during auditory verbal hallucinations in patients with schizophrenia [59], or during induced mental rumination [60]. Some authors also recently demonstrated that it is possible to discriminate inner speech content based on surface electromyography (EMG) signals with a median 92% accuracy [61]. However, other teams failed to obtain such results [32].

Many of these EMG studies concluded on the involvement of the speech motor system based on a difference in EMG amplitude by contrasting a period of inner speech production to a period of rest. However, as highlighted by [62], it is usually not enough to show an increase of speech muscle activity during inner speech to conclude that this activation is related to inner speech production. Indeed, three sorts of inference can be made based on the studies of electromyographic correlates of inner speech production, depending on the stringency of the control procedure. The stronger sort of inference is permitted by highlighting a discriminative pattern during covert speech production, as for instance when demonstrating a dissociation

between different speech muscles during the production of speech sounds of different phonemic class (e.g, contrasting labial versus non-labial words). According to [62], other (weaker) types of control procedures include i) comparing the EMG activity during covert speech production to a baseline period (without contrasting phonemic classes in covert speech utterances), or ii) comparing the activity of speech-related and non-speech related (e.g., forearm) muscle activity. Ideally, these controls can be combined by recording and contrasting speech and non-speech related muscles in different conditions (e.g., rest, covert speech, overt speech) of pronunciation of different speech sounds classes (e.g., labial versus non-labial).

Previous research studies carried out using the preferred procedure recommended by [62] suggest a discriminative patterns of electromyographic correlates according to the phonemic class of the words being covertly uttered [30, 31], which would corroborate the *motor simulation view* of inner speech. However, these studies used limited sample sizes (often less than ten participants) and worked mostly with children. These factors limit the generalisability of the above findings because i) low-powered experiments provide biased estimates of effects, ii) following the natural internalisation process, inner speech muscular correlates are expected to weaken with age and iii) a higher sensitivity could be attained by using modern sensors and signal processing methods.

The present study intends to bring new information to the debate between the *motor simulation view* and the *abstraction view* of inner speech, by focusing on an expanded form of inner speech: wilful nonword covert production. This work can be seen as a replication and extension of previous works carried out by McGuigan and collaborators [30, 31]. We aimed to demonstrate similar dissociations by using surface electromyography recorded over the lip (*orbicularis oris inferior*, OOI) and the *zygomaticus major* (ZYG) muscles. More precisely, given that rounded phonemes (such as /u/) are articulated with orbicular labial contraction, whereas spread phonemes (such as /i/) are produced with zygomaticus contraction, if the *motor simulation view* is correct, we should observe a higher average EMG amplitude recorded over the OOI during both the overt and inner production of rounded nonwords in comparison to spread nonwords. Conversely, we would expect a lower average EMG amplitude recorded over the ZYG during both the inner and overt production of rounded nonwords in comparison to spread nonwords. In addition, we would not expect to observe content-specific differences in EMG amplitude concerning the non speech-related muscles (i.e., forehead and forearm muscles).

## Methods

In the *Methods* and *Data analysis* sections, we report how we determined our sample size, all data exclusions, all manipulations, and all measures in the study [63]. A pre-registered version of our protocol can be found at: https://osf.io/czer4/.

### Participants

As previous studies of the electromyographic correlates of inner speech were mostly carried out with samples of children or young adults, used different kinds of EMG measures (surface EMG or needle EMG), and different kinds of signal processing methods, it was impractical to determine the effect size of interest for the current study. Therefore, we used sequential testing as our sampling procedure, based on the method described in [64] and [65]. We fixed a statistical threshold to $BF_{10} = 10$ and $BF_{10} = 1/10$ (i.e., $BF_{01} = 10$), testing the difference between the inner production of labial items versus the inner production of non-labial items on the standardised EMG amplitude of the lower lip (*orbicularis oris inferior*). In order to prevent potential experimenter and demand biases during sequential testing, the experimenter was blind to

BFs computed on previous participants [66]. All statistical analyses have been automatised and a single instruction was returned to the experimenter (i.e., "keep recruiting participants" or "stop the recruitment"). We fixed the maximum sample size to 100 participants.

As a result of the above sampling procedure, a total of 25 French-speaking female undergraduate students in Psychology from the Univ. Grenoble Alpes (mean age = 19.57, SD = 1.1). took part in this experiment, in exchange for course credits. It should be noted that this procedure did not work optimally because we later spotted an error in the EMG signal processing workflow. Thus, the sequential testing stopped earlier than it should have. These participants were recruited via mailing list, online student groups, and posters. Each participant provided a written consent and the present study was approved by the local ethics committee (Grenoble CERNI agreement #2016-05-31-9).

## Material

**EMG recordings.** EMG activity was recorded using TrignoTM Mini sensors (Delsys Inc.) with a sampling rate of 1926 samples/s, a band pass of 20 Hz (12 dB/oct) to 450 Hz (24 dB/oct) and were amplified by a TrignoTM 16-channel wireless EMG system (Delsys Inc.). These sensors consist of two 5 mm long, 1 mm wide parallel bars, spaced by 10 mm, which were attached to the skin using double-sided adhesive interfaces. The skin was cleaned by scrubbing it with 70% isopropyl alcohol. EMG signals were synchronised using the PowerLab 16/35 (ADInstrument, PL3516). Raw data from the EMG sensors were then resampled at a rate of 1 kHz and stored in digital format using Labchart 8 software (ADInstrument, MLU60/8).

EMG sensors were positioned over five muscles: the *corrugator supercilii* (COR), the *frontalis* (FRO), the *zygomaticus major* (ZYG), the *orbicularis oris inferior* (OOI), and the *flexor carpi radialis* (FCR). Given that the activity of the *orbicularis oris inferior* and *orbicularis oris superior* muscles has previously been observed to be strongly correlated and that the activity of the OOI was more strongly affected by the experimental manipulation [59, 60], we decided to record only the activity of the OOI in this study. The two speech-related muscles (OOI and ZYG) were chosen to show speech-specific EMG correlates, whereas the two non-speech related facial muscles (COR and FRO) were chosen to control for overall facial muscular activity. We also recorded the activity of the FCR of the non-dominant forearm to control for overall (body) muscular activity.

As reviewed in [67], the dominant side of the face displays larger movements than the left side during speech production, whereas the non-dominant side is more emotionally expressive. Therefore, we recorded the activity of control and emotion-linked muscles (i.e., COR and FRO) that were positioned on the non-dominant side of the face (i.e., the left side for right-handed participants), while sensors recording the activity of the speech muscles (i.e., ZYG and OOI) were positioned on the dominant side of the face.

The experiment was video-monitored using a Sony HDR-CX240E video camera to track any visible facial movements. A microphone was placed 20–30 cm away from the participant's lips to record any faint vocal production during the inner speech and listening conditions. Stimuli were displayed using the OpenSesame software [68] on a 19-inch colour monitor.

**Linguistic material.** We selected ten rounded and ten spread bi-syllabic nonwords (cf. Table 1). Each class of nonwords was specifically designed to either induce a greater activation of the lip muscle (rounded items) or a greater activation of the zygomaticus muscle (spread items). These stimuli were selected based on phonetic theoretical constraints, with the aim of maximising the differences between the two classes of non-words in their involvement of either the OOI or the ZYG muscle. More precisely, rounded items consisted in the repetition of a syllable containing a bilabial consonant followed by a rounded vowel, whereas spread

**Table 1. List of bisyllabic nonwords used in the test session.**

| rounded items | spread items |
|:---:|:---:|
| /mumu/ | /gigi/ |
| /pupu/ | /sese/ |
| /fofo/ | /lele/ |
| /mymy/ | /sisi |
| /pypy/ | /didi/ |
| /byby/ | /nini/ |
| /vøvø/ | /ʒiʒi/* |
| /pøpø/ | /lili/ |
| /bøbø/ | /ʁiʁi/ |
| /m̃ɔm̃ɔ/ | /gege/ |

*Because the production of the French palato-alveolar fricative in /ʒiʒi/ may involve a protrusion of the lips, this item theoretically slightly deviates from other items of this class.

items consisted in the repetition of a syllable containing a lingual consonant followed by a spread vowel.

## Procedure

Participants were seated in front of a computer screen while audio stimuli (when applicable) were presented through speakers on both sides of the screen. A video camera was positioned on one side of the screen to monitor facial movements. A microphone was positioned at approximately 10 cm of the participant to record possible speech sounds. After positioning of the EMG sensors, each participant underwent a relaxation session aiming to minimise pre-existing inter-individual variability on facial muscle contraction (approximate duration was 330 s). This relaxation session was recorded by a trained professional sophrology therapist. Baseline EMG measurements were performed during the last minute of this relaxation session, resulting in 60 s of EMG signal at baseline. By using this relaxation period as a baseline, we made sure that participants were all in a comparable relaxed state. In addition, several previous EMG studies have argued for the use of a relaxation period as a baseline, since mere resting periods may include some inner speech production (e.g., [69, 70], for a review).

Subsequently, participants went through a training session, during which they could get familiar with the main task. They trained with 8 stimuli in total (4 rounded nonwords and 4 spread nonwords, cf. S1 Text). Each training stimulus appeared in three conditions (for all participants): overt speech, inner speech and listening. Nonwords to be produced (covertly or overtly) were visually presented on the screen. Then, a central fixation dot appeared on the screen, indicating to the participant that s•he should utter the nonword (either overtly or covertly). This aimed to ensure that participants were actually producing a nonword, not just simply visually scanning it. In the overt speech condition, participants were asked to produce the nonword "just after the word disappeared from the screen", with "the most neutral intonation possible". In the inner speech condition, participants were asked to "innerly produce the nonword" (cf. the S1 Text for precise instructions in French), with "the most neutral intonation possible" and while remaining as still as possible. In the listening condition, the order of these two screens was reversed. A fixation dot was first presented (for 1 second), followed by a blank screen (for 1 second). The audio stimulus was presented when the blank screen appeared, while participants were asked to remain as still as possible.

After the training, participants moved to the experimental part, that included a novel list of 20 nonwords (cf. Table 1). Each nonword was presented 6 times in each condition for each participant. The EMG activity was recorded during the entire experiment. The periods of interest consisted in one-second portions, after each stimulus presentation and during either production or listening. This resulted in 60 observations (60 periods of 1 second) for both classes of nonword in each test condition. The total duration of the experiment ranged between 30 min and 40 min.

### EMG signal processing

EMG signal pre-processing was carried out using Matlab r2014a (Version 8.3.0.532, www. mathworks.fr). We first applied a 50Hz frequency comb filter to eliminate power noise. Then, we applied a 20 Hz—450 Hz bandpass filter to the EMG signals, in order to focus on the 20–450 Hz frequency band, following current recommendations for facial EMG studies [71, 72].

Although participants were explicitly asked to remain still during inner speech production or listening, small facial movements (such as swallowing movements) sometimes occurred. Such periods were excluded from the final sample of EMG signals. To remove these signals, we first divided the portions of signals of interest into periods of 1 second. The baseline condition was therefore composed of 60 trials of 1 second. The periods of interest in all the speech conditions consisted of the 1-second interval during which the participants either produced speech or listened to speech. It is possible that the nonword took less than 1 second to be produced, but since there was no way to track when production started and ended in the inner speech condition, the entire 1-second period was kept. Therefore, the overt speech condition was composed of 6 repetitions of each nonword, that is 6x20 trials of 1 second. The "inner speech" and "listening" conditions were similarly composed of 6x20 trials of 1 second. Then, we visually inspected the EMG signals recorded during each trial and listened to the audio signal simultaneously recorded. In all conditions, any time a non-speech noise (such as coughing or yawning) was audible in a trial, the trial was discarded (i.e., we did not include this trial in the final analysis, for any of the recorded muscles). In the listening and overt speech conditions, if a burst of EMG activity was present after the relevant audio speech signal, then the trial was discarded. In the overt speech condition, if the participant started too early or too late and only part of the nonword was recorded in the audio signal, then the corresponding trial was discarded. In the inner speech and listening condition, if a large EMG burst of activity was present, potentially associated with irrelevant non-speech activity, we excluded the trial. The fact that the artefact rejection procedures slightly differ in the various conditions is not an issue since we do not directly compare between conditions. Instead, we compare the EMG correlates of the two classes of nonwords within each condition.

This inspection was realised independently by two judges. Subsequently, we only kept the trials that were not rejected by any of these two judges (i.e., we removed a trial as soon as it was rejected by at least one judge). The agreement rate between the two judges was of 87.82% (with a moderate Cohen's $\kappa$ of approximately 0.48). The overall procedure led to an average (averaged over participants) rejection rate of 22.96% (SD = 6.49) in the baseline condition and 18.49% (SD = 6.48) in the other conditions.

After pre-processing and artefact rejection, we computed the by-trial average amplitude of the centered and rectified EMG signal. This provided a score for each muscle of interest (OOI, ZYG, FRO, COR, FCR) in each condition (Baseline, Overt Speech, Inner Speech, Listening) and for each participant. Absolute EMG values are not meaningful as muscle activation is never null, even in resting conditions, due in part to physiological noise. In addition, there are inter-individual variations in the amount of EMG activity in the baseline. To normalise and

standardise for baseline activity across participants, we thus expressed the EMG amplitude as a z-score from baseline activity (i.e., we subtracted the mean amplitude of the centred and rectified baseline signal and divided the result by the standard deviation of the centred and rectified baseline signal), thereafter referred to as $\delta$.

## Data analysis

Statistical analyses were conducted using R version 3.5.3 [73], and are reported with the papaja [74] and knitr [75] packages. To assess the effects of the condition and the class of nonwords on the standardised EMG amplitude, we analysed these data using *Condition* (3 modalities: speech, inner speech, and listening) and *Class* of nonwords (2 modalities, rounded and spread, contrast-coded) as within-subject categorical predictors, and the standardised EMG amplitude as a dependent variable in a multivariate (i.e., with multiple outcomes) Bayesian multilevel linear model (BMLM). An introduction to Bayesian statistics is outside the scope of this paper. However, the interested reader is referred to [76] for an introduction to Bayesian multilevel modelling using the brms package.

In order to take into account the dependencies between repeated observations by participant, we also included in this model a varying intercept by participant. Contrary to what we pre-registered, we used a multivariate model (instead of separate models by muscle). This allowed us to estimate the correlation between each pair of muscles. Models were fitted with the brms package [77] and using weakly informative priors (see the S1 Text for code details). Two Markov Chain Monte-Carlo (MCMC) were run for each model to approximate the posterior distribution, including each 5.000 iterations and a warmup of 2.000 iterations. Posterior convergence was assessed examining trace plots as well as the Gelman-Rubin statistic $\hat{R}$. Constant effect estimates were summarised via their posterior mean and 95% credible interval (CrI), where a credible interval can be considered as the Bayesian analogue of a classical confidence interval. When applicable, we also report Bayes factors (BFs), computed using the Savage-Dickey method, which consists in taking the ratio of the posterior density at the point of interest divided by the prior density at that point. These BFs can be interpreted as an updating factor, from prior knowledge (what we knew before seeing the data) to posterior knowledge (what we know after seeing the data).

## Results

The *Results* section is divided into two parts. First, we present results from confirmatory (pre-registered) analyses, aiming to test whether it is possible to dissociate the activity of the OOI and the ZYG during inner speech production, according to the content of inner speech (here, the class of nonword). More precisely, we expected an increased EMG activity of the OOI during the inner production of rounded nonwords in comparison to spread nonwords. Conversely, we expected elevated EMG activity of the ZYG during the inner production of spread nonwords in comparison to rounded nonwords. Second, we present results from exploratory (non-preregistered) analyses.

To foreshadow the results, we did not observe such a clear dissociation between the EMG activity of the OOI and the ZYG muscles, neither in the inner speech condition nor in the overt speech condition. Contrary to theoretical expectations based on phonetics and speech production theory [78–81], the activity of both muscles was of higher amplitude for the pronunciation of rounded nonwords (as compared to spread nonwords) during overt speech production. Additionally, the EMG amplitude on both muscles of interest was similar during the inner production (or listening) of the two classes of nonwords. However, in the exploratory analyses section, we report results from supervised machine learning algorithms (classification

using random forests), showing a reasonable accuracy to classify EMG signals according to the class of nonwords during overt speech production. This strategy was however unsuccessful for the inner speech and the listening conditions.

Before moving to the statistical results, we represent the distribution of the whole dataset, by class, by condition and by muscle for the two main muscles of interest (OOI and ZYG) in Fig 1. More precisely, the first row of this figure represents the distribution of the standardised EMG scores in the inner speech condition, the second row depicts the distribution of these scores in the listening condition, whereas the third row depicts the distribution of the standardised EMG scores in the overt speech condition. The first column depicts the distribution of the standardised EMG scores recorded over the OOI muscle whereas the second one represents the distribution of the standardised EMG scores recorded over the ZYG muscle. Each individual data point is represented as a vertical bar along the x-axis of each panel whereas the vertical coloured line represents the class-specific median. Additionally, a vertical dashed line is plotted at zero, which represents the baseline level. Thus, a positive value on the x-axis represents EMG standardised scores that are higher than baseline.

In Table 2, we report the mean standardised EMG amplitude of all recorded muscles in each condition. Given the skewness of the distribution of these scores, the mean and the standard deviation (SD) are not the best indicators of the central tendency and dispersion of these distributions. Therefore, we also report the median, the median absolute deviation (MAD), and the inter-quartile range (IQR).

We also created a `shiny` application [82] allowing for further visual exploration of the data by muscle, by condition, and by participant, in the 3D space formed by three (to be chosen) muscles. This application is available online (at https://barelysignificant.shinyapps.io/3d_plotly/) and the associated code is available in the OSF repository (https://osf.io/czer4).

## Confirmatory (preregistered) analyses

**Bayesian multivariate multilevel Gaussian model.**   We then compared the standardised EMG amplitude $\delta$ for each muscle in each condition (Overt Speech, Inner Speech, Listening) by fitting a multivariate multilevel Gaussian model (as detailed previously in the Methods section). We predicted a higher increase of OOI activity during the inner production of rounded items in comparison to spread items and conversely, a higher increase of ZYG activity during the inner production of rounded items in comparison to spread items. These predictions should also apply to the overt speech condition (and to the listening condition). We should not observe any by-class differences of FRO and COR activity in any condition.

The results of the Bayesian Gaussian multivariate model are summarised in Table 3. This table reports the estimated average EMG amplitude in each condition and the corresponding BF. As they are not the main focus of interest here and for the sake of clarity, descriptive results for the other two facial muscles and for the forearm muscle are reported in the S1 Text. This analysis revealed that the EMG amplitude of the OOI was higher than baseline (the standardised score was above zero) in every condition whereas it was only the case in the overt speech condition for the ZYG. Moreover, in all conditions, the EMG amplitude of the ZYG was lower than that of the OOI. Crucially, we did not observe the hypothesised difference according to the class of nonwords on the OOI during inner speech production ($\beta = 0.071$, 95% CrI [-0.204, 0.342], $BF_{01} = 64.447$) nor on the ZYG ($\beta = 0.005$, 95% CrI [-0.031, 0.041], $BF_{01} = 532.811$).

Fig 2 depicts these results by representing the distribution of the raw data (coloured dots) along with the predictions from this model. The black dots and vertical intervals represent the predicted mean and associated 95% credible interval for each class of non-word, each condition and for the OOI and the ZYG. Coherently with Table 3, this figure shows that the fitted

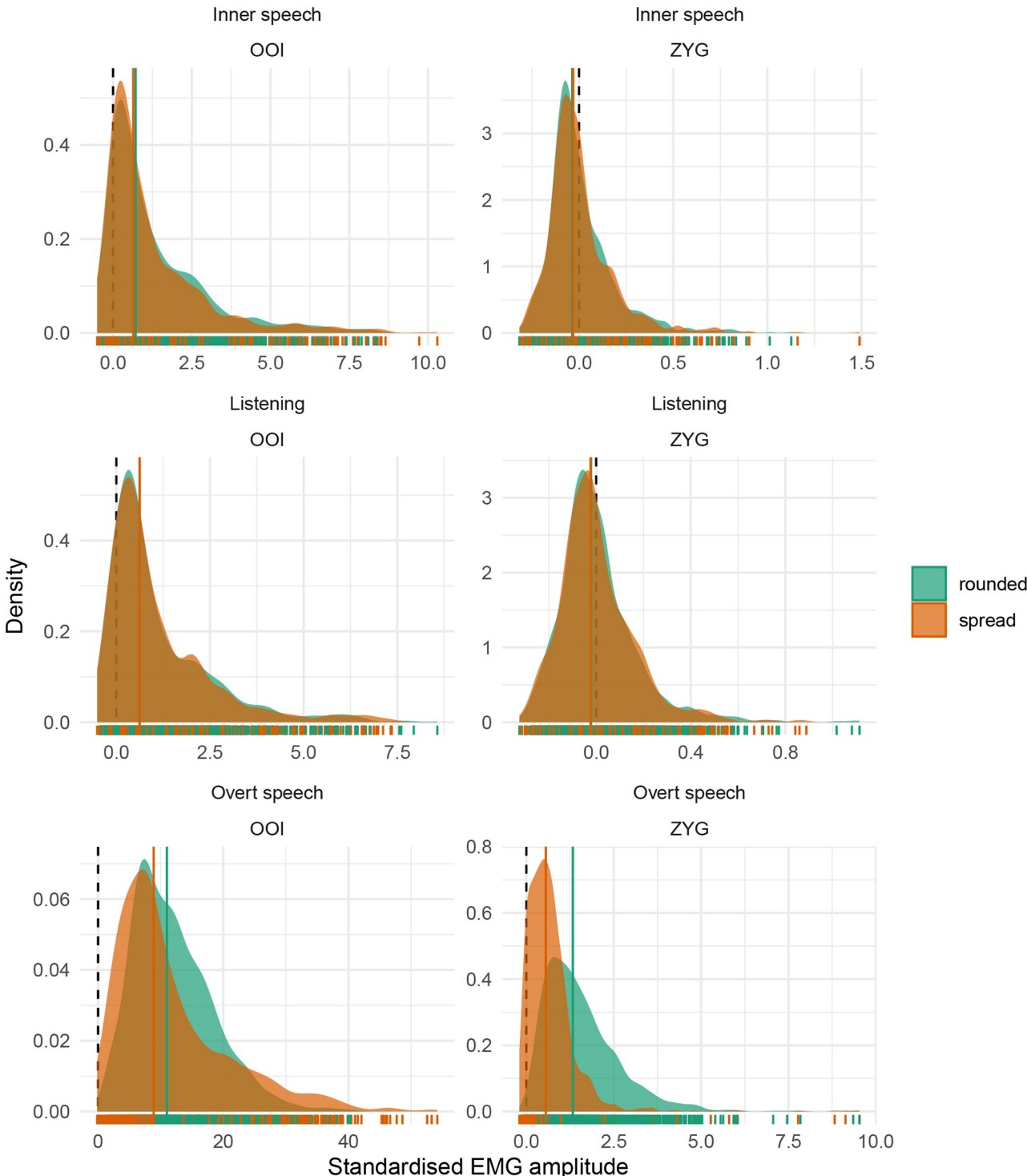

**Fig 1. Distribution of standardised EMG scores by class and by muscle.** The first row corresponds to the inner speech condition, the second one to the listening condition, and the third one to the overt speech condition. The first column depicts the EMG amplitude recorded over the OOI muscle while the second column represents the EMG amplitude recorded over the ZYG muscle. Each individual data point is represented as a vertical bar along the x-axis. The vertical coloured line represents the by-class median. Please note that the scale of the x-axis may differ considerably between panels.

**Table 2. Descriptive statistics of the standardised EMG amplitude for each muscle in each condition.**

| Condition | Item | Muscle | Mean | Median | SD | MAD | IQR |
|---|---|---|---|---|---|---|---|
| innerspeech | rounded | COR | 0.34 | 0.17 | 0.54 | 0.42 | 0.65 |
| innerspeech | spread | COR | 0.32 | 0.16 | 0.54 | 0.40 | 0.63 |
| listening | rounded | COR | 0.22 | 0.09 | 0.46 | 0.36 | 0.57 |
| listening | spread | COR | 0.24 | 0.11 | 0.46 | 0.37 | 0.58 |
| speech | rounded | COR | 0.29 | 0.17 | 0.52 | 0.40 | 0.56 |
| speech | spread | COR | 0.25 | 0.13 | 0.52 | 0.37 | 0.52 |
| innerspeech | rounded | FCR | -0.01 | -0.04 | 0.18 | 0.09 | 0.13 |
| innerspeech | spread | FCR | 0.00 | -0.04 | 0.19 | 0.09 | 0.12 |
| listening | rounded | FCR | -0.01 | -0.04 | 0.15 | 0.09 | 0.12 |
| listening | spread | FCR | 0.00 | -0.03 | 0.15 | 0.09 | 0.12 |
| speech | rounded | FCR | -0.01 | -0.03 | 0.14 | 0.08 | 0.11 |
| speech | spread | FCR | -0.01 | -0.03 | 0.13 | 0.09 | 0.12 |
| innerspeech | rounded | FRO | 0.63 | 0.47 | 0.68 | 0.66 | 0.95 |
| innerspeech | spread | FRO | 0.62 | 0.47 | 0.67 | 0.64 | 0.95 |
| listening | rounded | FRO | 0.59 | 0.40 | 0.67 | 0.57 | 0.82 |
| listening | spread | FRO | 0.60 | 0.41 | 0.67 | 0.58 | 0.85 |
| speech | rounded | FRO | 0.66 | 0.48 | 0.71 | 0.62 | 0.88 |
| speech | spread | FRO | 0.62 | 0.46 | 0.72 | 0.60 | 0.81 |
| innerspeech | rounded | OOI | 1.31 | 0.72 | 1.63 | 0.94 | 1.80 |
| innerspeech | spread | OOI | 1.24 | 0.64 | 1.67 | 0.85 | 1.61 |
| listening | rounded | OOI | 1.11 | 0.62 | 1.40 | 0.85 | 1.44 |
| listening | spread | OOI | 1.11 | 0.62 | 1.42 | 0.83 | 1.40 |
| speech | rounded | OOI | 12.01 | 10.99 | 6.44 | 6.22 | 8.75 |
| speech | spread | OOI | 11.71 | 8.88 | 9.13 | 6.56 | 10.25 |
| innerspeech | rounded | ZYG | 0.01 | -0.04 | 0.17 | 0.11 | 0.17 |
| innerspeech | spread | ZYG | 0.00 | -0.03 | 0.18 | 0.11 | 0.16 |
| listening | rounded | ZYG | 0.01 | -0.02 | 0.17 | 0.12 | 0.17 |
| listening | spread | ZYG | 0.00 | -0.02 | 0.16 | 0.12 | 0.18 |
| speech | rounded | ZYG | 1.57 | 1.33 | 1.10 | 0.92 | 1.29 |
| speech | spread | ZYG | 0.69 | 0.56 | 0.78 | 0.51 | 0.70 |

model predicts no noticeable differences between the two classes of non-words in any condition for the OOI muscle. However, it predicts a higher average EMG amplitude associated with the rounded item as compared to the spread items in the overt speech condition for the ZYG muscle.

Before proceeding further with the interpretation of the results, it is essential to check the quality of this first model. A useful diagnostic of the model's predictive abilities is known as *posterior predictive checking* (PPC) and consists in comparing observed data to data simulated from the posterior distribution [83]. The idea behind PPC is that a good model should be able to generate data that resemble the observed data [84]. In this vein, Fig 3 represents the distribution of the whole dataset (across all participants and conditions) by muscle (the dark blue line) along with the distribution of hypothetical datasets generated from the posterior distribution of the model (the light blue lines). As can be seen from this figure, the distributions of the data generated from the model differ considerably from the distribution of the observed data. Therefore, in the next section, we turn to a more appropriate model for these data.

**Bayesian multivariate multilevel distributional Skew-Normal model.** Fig 3 reveals an important failure of the first model, as it fails to generate data that look like the data we have

**Table 3. Estimates from the Gaussian BMLM concerning the OOI and the ZYG.**

| Response | Term | Estimate | SE | Lower | Upper | Rhat | BF01 |
|---|---|---|---|---|---|---|---|
| OOI | Inner Speech | 1.21 | 0.27 | 0.67 | 1.75 | 1.00 | 0.04 |
| OOI | Listening | 1.09 | 0.23 | 0.65 | 1.53 | 1.00 | <0.001 |
| OOI | Overt Speech | 11.59 | 1.28 | 9.08 | 14.18 | 1.00 | <0.001 |
| OOI | Inner Speech x Class | 0.07 | 0.14 | -0.20 | 0.34 | 1.00 | 64.45 |
| OOI | Listening x Class | -0.08 | 0.20 | -0.47 | 0.32 | 1.00 | 47.05 |
| OOI | Overt Speech x Class | 0.02 | 0.19 | -0.35 | 0.39 | 1.00 | 52.11 |
| ZYG | Inner Speech | 0.00 | 0.03 | -0.05 | 0.06 | 1.00 | 379.5 |
| ZYG | Listening | 0.01 | 0.02 | -0.04 | 0.05 | 1.00 | 388.4 |
| ZYG | Overt Speech | 1.15 | 0.15 | 0.86 | 1.43 | 1.00 | <0.001 |
| ZYG | Inner Speech x Class | 0.01 | 0.02 | -0.03 | 0.04 | 1.00 | 532.81 |
| ZYG | Listening x Class | 0.00 | 0.03 | -0.06 | 0.05 | 1.00 | 389.12 |
| ZYG | Overt Speech x Class | 0.86 | 0.03 | 0.81 | 0.91 | 1.00 | <0.001 |

For each muscle (response), the first three lines represent the estimated average amplitude in each condition, and its standard error (SE). The three subsequent rows represent the estimated average difference between the two classes of nonwords in each condition (i.e., the interaction effect). The 'Lower' and 'Upper' columns contain the lower and upper bounds of the 95% CrI, whereas the 'Rhat' column reports the Gelman-Rubin statistic. The last column reports the Bayes factor in favour of the null hypothesis (BF01).

collected. More precisely, the collected data look right-skewed, as it usually happens with physiological measurements. To improve on the Gaussian model, we then assumed a Skew-normal distribution for the response variable (the standardised EMG amplitude $\delta$). The Skew-normal distribution is a generalisation of the Gaussian distribution with three parameters $\xi$ (xi), $\omega$ (omega), and $\alpha$ (alpha) for location, scale, and shape (skewness), respectively (note that the Gaussian distribution can be considered a special case of the Skew-normal distribution when $\alpha = 1$). In addition, we also improved the first model by turning it into a *distributional model*, that is, a model in which we can specify predictor terms for all parameters of the assumed response distribution [85]. More precisely, we used this approach to predict both the location, the scale, and the skewness of the Skew-Normal distribution (whereas the first model only allowed predicting the mean of a Gaussian distribution). As can been seen in Fig 4, this second model seems better than the first one at generating data that fit the observed data.

The estimates of this second model are summarised in Table 4 and Fig 5. According to this model, the EMG amplitude of the OOI was higher than baseline (the estimated standardised score was above zero) in every condition whereas, for the ZYG, it was only the case in the overt speech condition. We did not observe the hypothesised difference according to the class of nonwords during inner speech production, neither on the OOI ($\beta = 0.025$, 95% CrI [-0.012, 0.064], $BF_{01} = 64.447$) nor on the ZYG ($\beta = 0.004$, 95% CrI [-0.007, 0.014], $BF_{01} = 532.811$).

Predictions from this model are visually represented in Fig 5. This figure differs from Fig 2 (showing the predictions of the Gaussian model) in that the second model (the Skew-normal model) predicts shifts in location for both the OOI and the ZYG muscles according to the class of non-word in overt speech prediction. In contrast, the first model (the Gaussian model) predicted a by-class difference only for the ZYG muscle.

## Exploratory (non-preregistered) analyses

In the previous section, we tried to predict the average EMG amplitude by condition on each single muscle. Although this approach was appropriate to tackle our initial research question (i.e., can we distinguish muscle-specific EMG correlates of inner speech production?), it is not

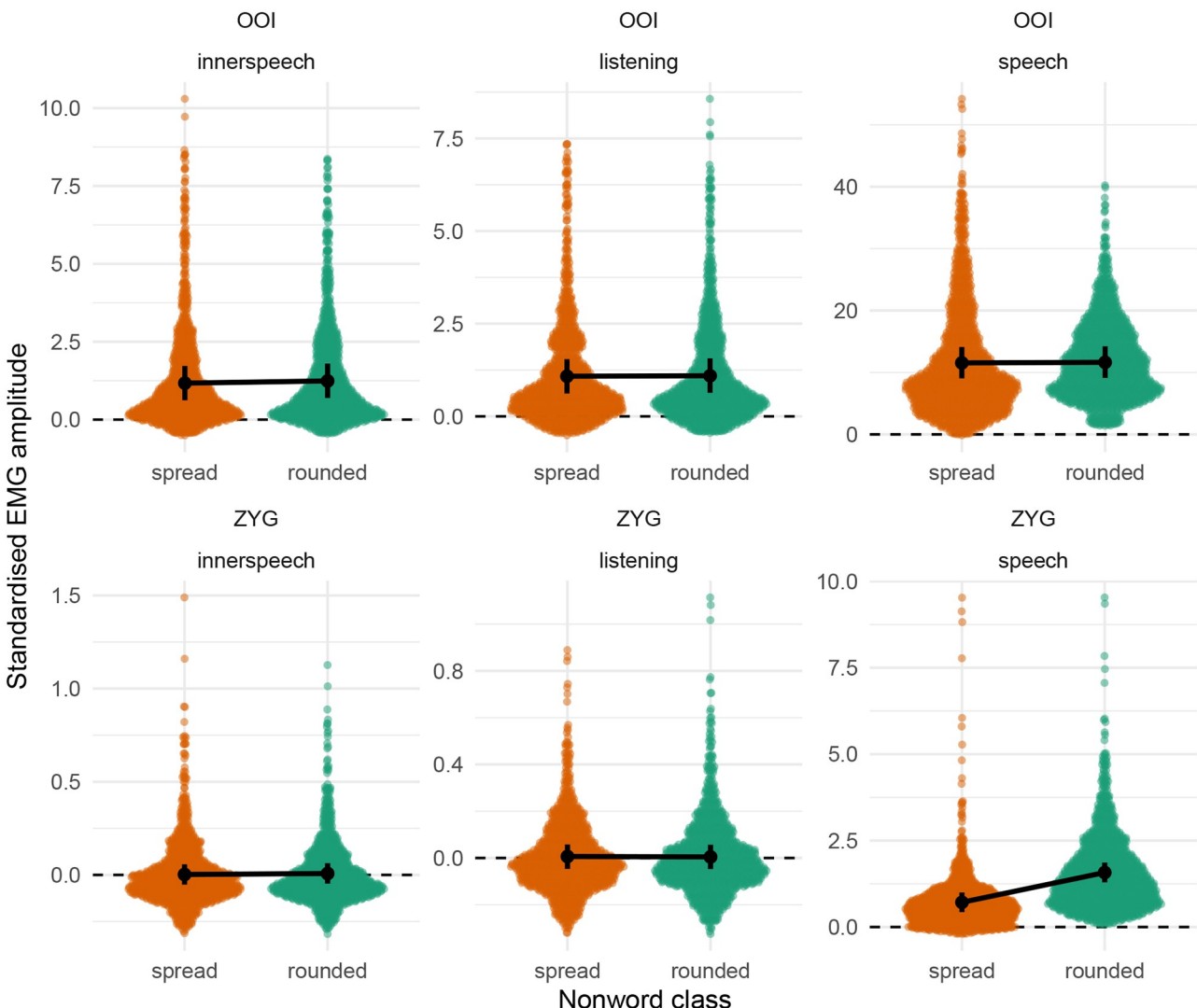

**Fig 2. Raw data along with posterior predictions of the first model for the OOI and the ZYG muscles.** Dots represent the mean prediction of this model by condition, whereas the vertical error bars represent the 95% credible intervals around the mean.

optimal to answer more general questions such as "can we predict the content of inner speech based on the available EMG data?". In Fig 6, we depict the distribution of the by-word averaged EMG scores in the 2D space formed by the OOI and the ZYG muscles. This figure reveals that although different nonwords produced in overt speech seem difficult to discriminate on the basis of a single muscle (cf. Fig 1), it seems easier to discriminate them in the space formed by two muscles (here OOI and ZYG). More precisely, the two classes of nonwords seem to form two separate clusters in the overt speech condition, but these clusters do not seem discriminable in the inner speech or in the listening condition.

In other words, it is easier to discriminate these signals in the multidimensional space of all speech muscles, rather than by considering each muscle independently. Thus, we used a supervised machine learning algorithm aiming to classify speech signals according to the class of nonwords. Broadly, the machine learning approach seeks to find a relationship between an input $X$ (e.g., EMG recordings over the four facial muscles) and an output $Y$ (e.g., the class of

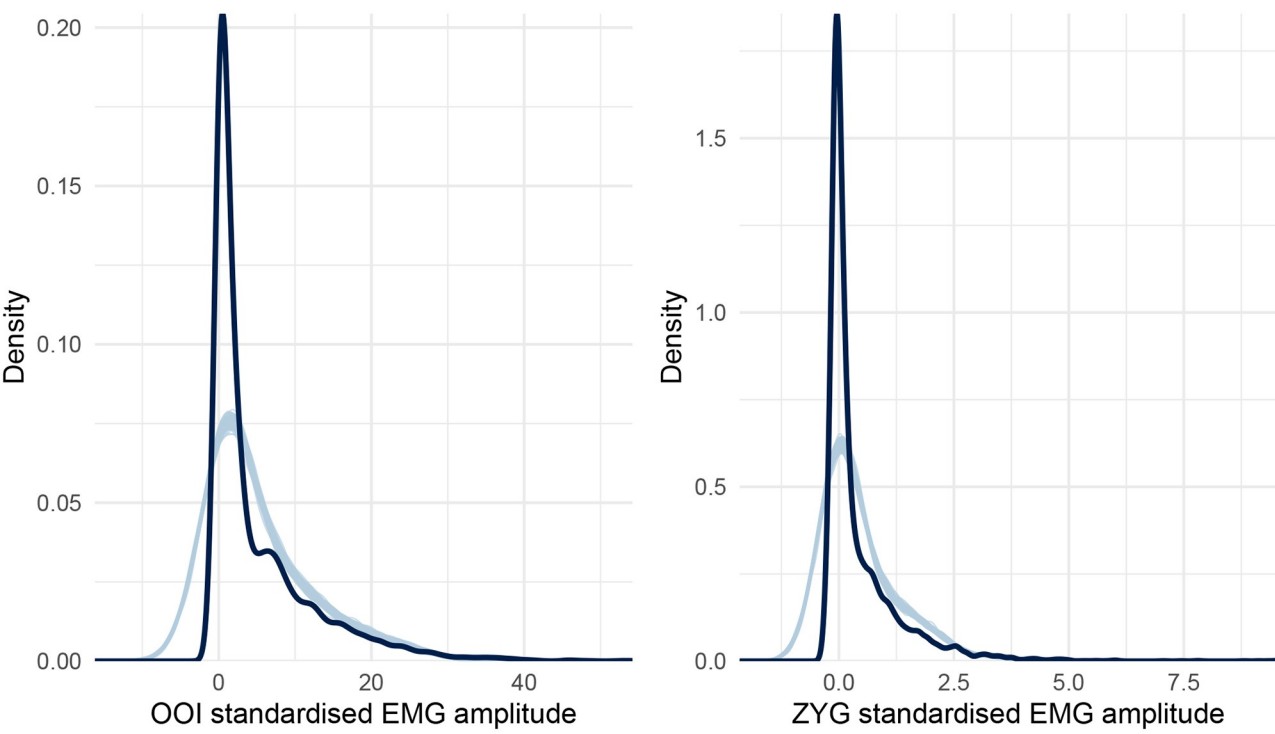

**Fig 3. Posterior predictive checking for the first model concerning the OOI and ZYG muscles.** The dark blue line represents the distribution of the raw data (across all conditions) whereas light blue lines are dataset generated from the posterior distribution.

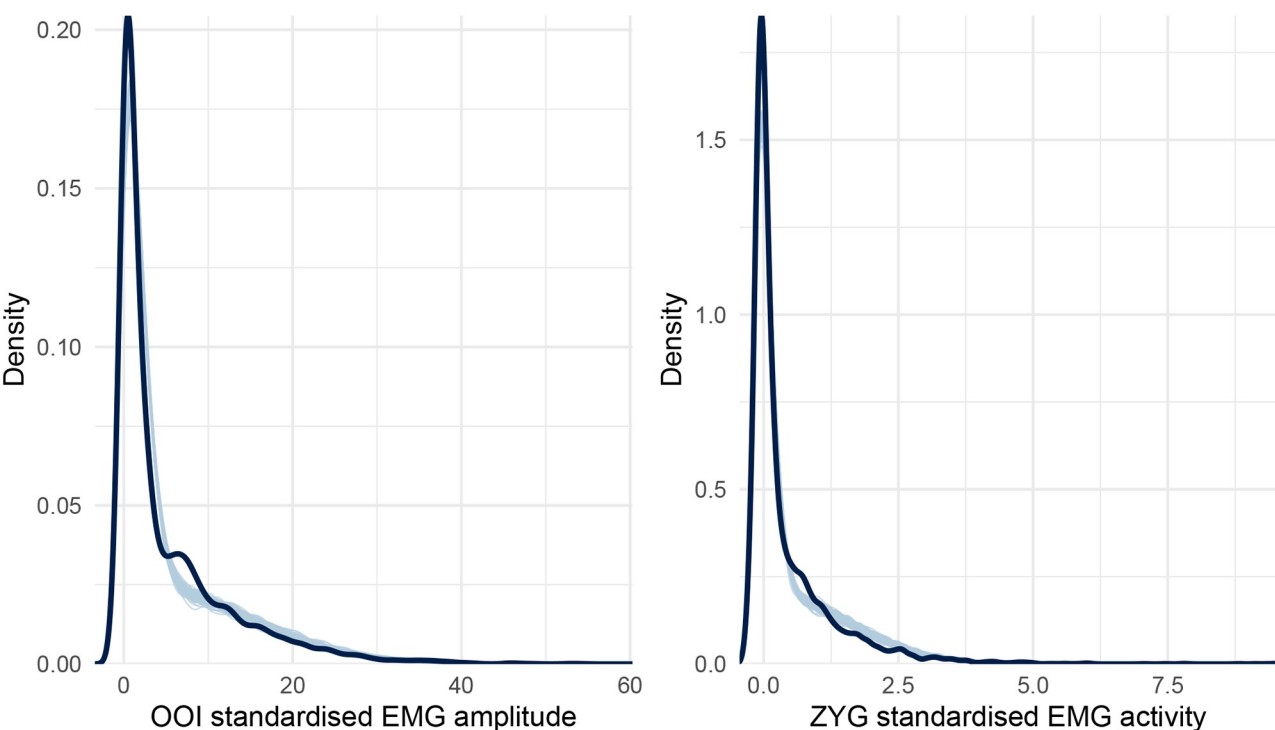

**Fig 4. Posterior predictive checking for the Skew-Normal model concerning the OOI and ZYG muscles.** The dark blue line represents the distribution of the raw data whereas light blue lines are dataset generated from the posterior distribution.

**Table 4. Estimates from the distributional Skew-Normal model concerning the OOI and the ZYG.**

| Response | Term | Estimate | SE | Lower | Upper | Rhat | BF01 |
|----------|------|----------|-----|-------|-------|------|------|
| OOI | Inner Speech | 1.47 | 0.03 | 1.41 | 1.53 | 1.00 | 0.04 |
| OOI | Listening | 1.24 | 0.02 | 1.19 | 1.29 | 1.00 | <0.001 |
| OOI | Overt Speech | 12.15 | 0.14 | 11.87 | 12.43 | 1.00 | <0.001 |
| OOI | Inner Speech x Class | 0.03 | 0.02 | -0.01 | 0.06 | 1.00 | 64.45 |
| OOI | Listening x Class | 0.00 | 0.02 | -0.05 | 0.05 | 1.00 | 47.05 |
| OOI | Overt Speech x Class | 1.42 | 0.18 | 1.05 | 1.78 | 1.00 | 52.11 |
| ZYG | Inner Speech | 0.02 | 0.00 | 0.01 | 0.02 | 1.00 | 379.5 |
| ZYG | Listening | 0.01 | 0.00 | 0.00 | 0.02 | 1.00 | 388.4 |
| ZYG | Overt Speech | 1.21 | 0.02 | 1.18 | 1.24 | 1.00 | <0.001 |
| ZYG | Inner Speech x Class | 0.00 | 0.01 | -0.01 | 0.01 | 1.00 | 532.81 |
| ZYG | Listening x Class | 0.00 | 0.01 | -0.02 | 0.02 | 1.00 | 389.12 |
| ZYG | Overt Speech x Class | 0.39 | 0.02 | 0.35 | 0.43 | 1.00 | <0.001 |

For each muscle (response), the first three lines represent the estimated average amplitude in each condition, and its standard error (SE). The three subsequent rows represent the estimated average difference between the two classes of nonwords in each condition (i.e., the interaction effect). The 'Lower' and 'Upper' columns contain the lower and upper bounds of the 95% CrI, whereas the 'Rhat' column reports the Gelman-Rubin statistic. The last column reports the Bayes factor in favour of the null hypothesis (BF01).

nonwords). Once trained, it allows predicting a value of the output based on some input values, whose prediction can be evaluated against new observations.

We used a random forest algorithm, as implemented in the `caret` package [86]. Random forests (RFs) represent an ensemble of many decision trees (a forest), which allow predictions to be made based on a series of decision rules (e.g., is the score on predictor $x_1$ higher or lower than $z$? If yes, then . . ., if not, then . . .). The specificity of RFs is to combine a large number of trees (usually above 100 trees), and to base the final conclusion on the average of these trees, thus preventing overfitting. We used three separate RFs to classify EMG signals in each condition (Overt Speech, Inner Speech, and Listening).

To evaluate the performance of this approach, we report the raw accuracy (along with its resampling-based 95% confidence interval), or the proportion of data points in the test dataset for which the RF algorithm predicted the correct class of nonwords. First, we randomly split the entire dataset into a training (80%) and a test set (20%). The training set was used for the learning whereas the test set was used to evaluate the predictions of the algorithm. To prevent overfitting, we used repeated 10-fold cross-validation during the learning phase.

**Predicting the class of nonwords during overt speech production.** We first tried to predict the class of nonwords produced in overt speech, based on the activity of the four facial muscles (OOI, ZYG, COR, FRO). Each predictor was centred to its mean and standardised before the analysis.

This analysis revealed an overall classification accuracy of 0.847, 95% CI [0.814, 0.876] (cf. confusion matrix in Table 5). Examining the relative importance of each feature (i.e., each muscle) for prediction revealed that the muscles containing most information to discriminate the two classes of nonwords were the ZYG and the OOI, whereas, as predicted, forehead muscles did not seem to strongly contribute to predictive accuracy in the overt speech condition.

**Predicting the class of nonwords during inner speech production and listening.** We then applied the same strategy (the same algorithm) to the signals recorded in the inner speech and listening conditions. The results of these analyses are reported in Tables 6 and 7.

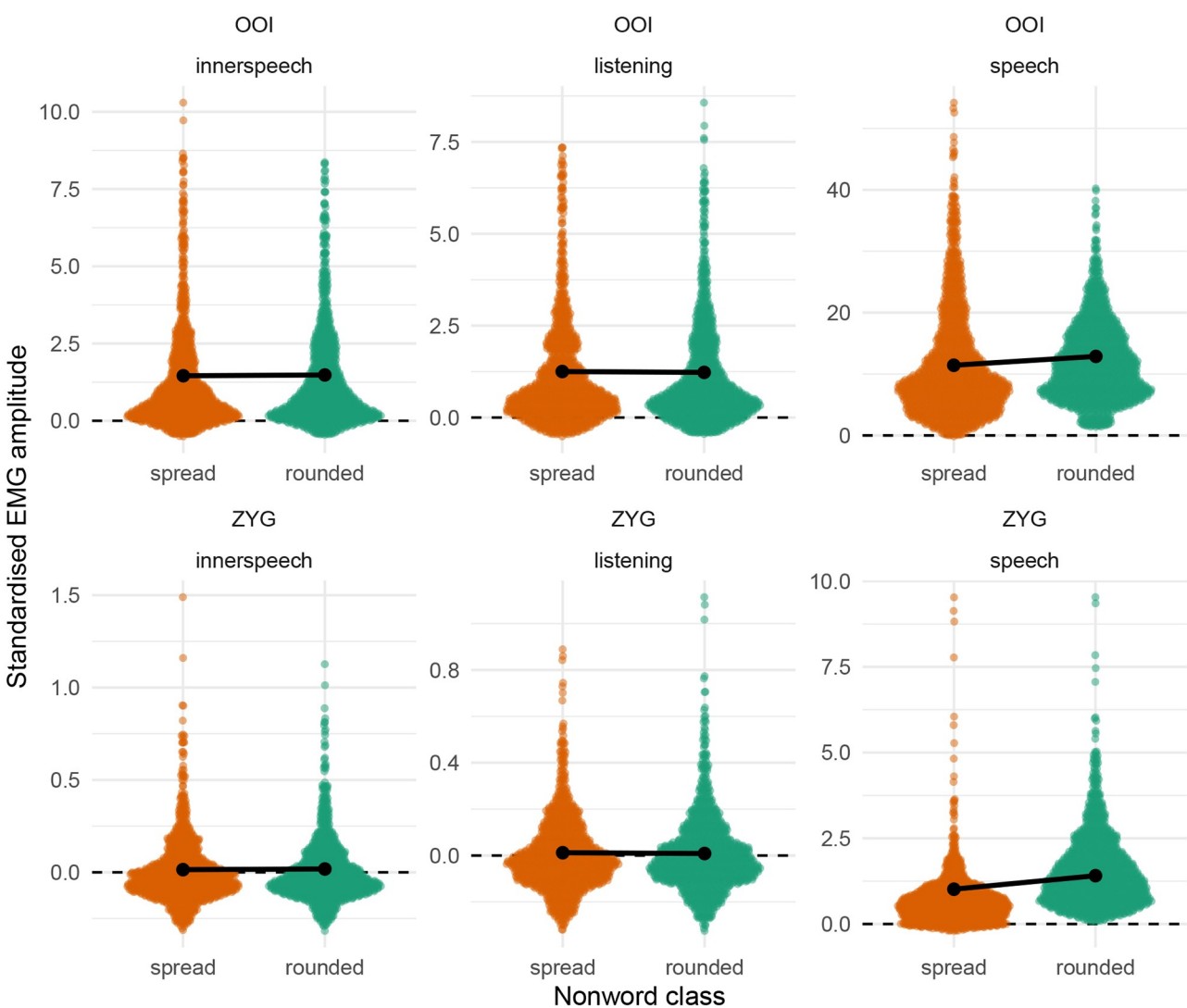

**Fig 5. Raw data along with posterior predictions of the third model for the OOI and the ZYG muscles.** Dots represent the mean prediction of this model by condition (concerning the location parameter) whereas the vertical error bars represent the 95% credible intervals.

This analysis revealed an overall classification accuracy of 0.472, 95% CI [0.426, 0.52] in the inner speech condition, which indicates that the RF algorithm did not allow discriminating the two classes of nonwords better than random guessing. As the classification accuracy in the inner speech and listening conditions was not better than chance, we do not report the relative importance of the predictors. Indeed, it would be difficult to interpret the importance of predictors for a classification task at which they do not perform better than chance.

This analysis similarly revealed an overall classification accuracy of 0.46, 95% CI [0.413, 0.507] in the listening condition.

## Discussion

In the present study we aimed to replicate and extend previous findings showing that facial electromyography can be used to discriminate expanded inner speech content [30, 31]. As these studies used small samples of children, our study aimed to examine whether such results

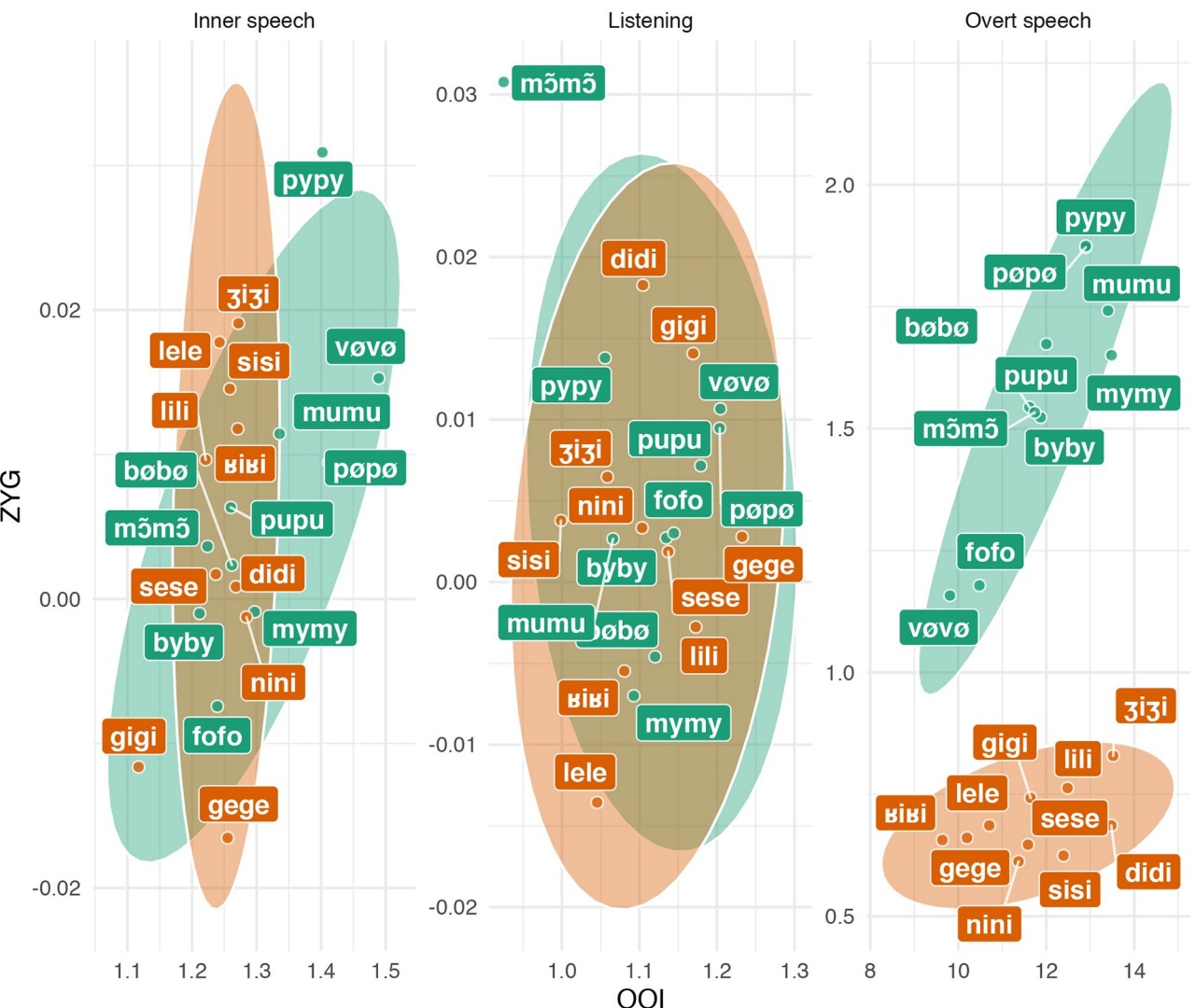

**Fig 6. Average standardised EMG amplitude for each nonword in each condition, in the 2D space formed by the OOI and the ZYG.** Ellipses represent 95% data ellipses, that is, the 95% contours of a bivariate normal distribution.

can be reproduced using surface electromyography and modern signal processing methods in an adult sample.

To this end, it was crucial to first show that the EMG correlates of our two classes of non-words were discriminable during overt speech production. Surprisingly, the data we collected during overt speech production do not corroborate the hypothesis according to which the average EMG amplitude of the OOI should be higher during the production of "rounded"

**Table 5. Confusion matrix with by-class error for the overt speech condition.**

| Prediction | Reference | | class.error |
|---|---|---|---|
| | **rounded** | **spread** | |
| rounded | 917 | 163 | 0.151 |
| spread | 198 | 898 | 0.181 |

**Table 6. Confusion matrix with by-class classification error for the inner speech condition.**

| Prediction | Reference | | class.error |
|---|---|---|---|
| | **rounded** | **spread** | |
| rounded | 386 | 502 | 0.565 |
| spread | 473 | 454 | 0.510 |

**Table 7. Confusion matrix with by-class classification error for the listening condition.**

| Prediction | Reference | | class.error |
|---|---|---|---|
| | **rounded** | **spread** | |
| rounded | 426 | 499 | 0.539 |
| spread | 508 | 406 | 0.556 |

nonwords as compared to "spread" nonwords. For both orofacial speech muscles (OOI and ZYG), the average EMG amplitude was higher for rounded nonwords than for spread nonwords during overt speech production. Moreover, whereas the average EMG amplitude recorded over speech muscles was higher than baseline in both the inner speech and listening conditions, we did not find differences of activation according to the content of the material (the class of nonword). An automatic classification approach, using the four facial muscles (OOI, ZYG, COR, FRO), revealed that although it was possible to discriminate EMG signals related to the two classes of nonwords with a reasonable accuracy during overt speech production, this approach failed in discriminating these two classes during inner speech production or during listening. We also observed a higher EMG amplitude recorded over the facial (both orofacial and non-orofacial) muscles during inner speech production and during the listening of speech production than during rest. However, as pinpointed by [62], this observation is not sufficient to conclude that these activations were actually related to inner speech production, because i) both orofacial speech-related muscles and forehead non-speech related muscles showed similar EMG amplitude changes from baseline and ii) we did not observe different changes in EMG amplitude depending on the content of inner speech (i.e., depending on the class of nonword to be uttered).

Before discussing the theoretical implications of these results, two main issues are worth discussing. First, how can we explain that rounded nonwords were associated with higher EMG amplitude during overt speech on both OOI and ZYG muscles? Second, how can we explain the indiscriminability of inner speech content, which seems to contradict classic as well as recent findings in the field [61]? We turn to each of these questions in the following.

To answer the first question, we began by comparing our results to results obtained by another group [87]. The authors of this study recorded surface EMG activity from five participants while they were producing seven facial expressions and five isolated vowel sounds (/a/, /e/, /i/, /o/, /u/), repeated five times each. They recorded EMG activity over eight facial muscles (the zygomaticus major (ZYG), the risorius (RIS), the orbicularis oris superior (OOS) and inferior (OOI), the mentalis (MEN), the depressor anguli oris (DAO), the levator labii superioris (LLS) muscles, and the digastric muscle (DIG)). We divided these vowels in two classes to fit our own classes of nonwords. More precisely, we have created the following two classes: a *rounded* class, composed of the vowels /o/ and /u/, and a *spread* class, composed of the vowels /e/ and /i/ (note that we did not include the vowel /a/ because it theoretically does not fit in one of these two categories). We present the average EMG amplitude recorded over the OOI and the ZYG according to the vowel class in Table 8.

**Table 8. Standardised EMG amplitude recorded over the OOI and the ZYG during overt speech production of rounded versus spread vowels in Eskes et al. (2017).**

| Muscle | Item | Observations | Mean | SD | Median | Histogram |
|--------|------|-------------|------|-----|--------|-----------|
| OOI | rounded | 50 | 59.70 | 60.09 | 42.03 | |
| OOI | spread | 50 | 22.15 | 11.92 | 20.65 | |
| ZYG | rounded | 50 | 7.39 | 3.78 | 6.27 | |
| ZYG | spread | 50 | 10.15 | 6.20 | 7.99 | |

The number of observations is given by the number of vowels to be pronounced in each category (2) times the number of repetitions (5) times the number of participants (5), for a total of 50 observations per cell.

We notice that [87] have indeed observed the dissociation we initially predicted, that is, that the EMG amplitude recorded over the OOI was higher during the pronunciation of rounded vowels than during pronunciation of spread vowels, whereas the reverse pattern was observed concerning the ZYG. Paired-samples Wilcoxon signed rank tests revealed a shift in location (pseudomedian) between rounded and spread items for the OOI ($\beta$ = 24.12, 95% CI [15.19, 40.77], V = 1184, p <.001) with rounded items being associated with a higher location than spread items. This analysis also revealed a shift in the inverse direction concerning the ZYG ($\beta$ = -1.51, 95% CI [-2.94, -0.48], V = 275, p <.001). However, one crucial difference between [87] design and ours is the complexity of the linguistic material. Whereas [87] used single phonemes, we chose to use bisyllabic nonwords to increase the ecological validity of the paradigm. Although these nonwords were specifically created to theoretically increase the engagement of either the OOI or the ZYG (cf. the "Linguistic material" section), it is reasonable to expect differences in the average EMG patterns between isolated phonemes and nonwords. More precisely, we expect the *average* EMG amplitude associated with the production of a given phoneme (e.g., /y/) to be impacted by the production of the consonant (e.g., /b/) it is paired with, due to coarticulation. More generally, we could hypothesise that the difference between the *average* EMG amplitude recorded during the production of the phoneme /i/ and during the production of the phoneme /y/ could be reduced when these phonemes are coarticulated in CV or CVCV sequences like /byby/ or /didi/ (as in our study). In other words, we might expect an interaction effect between the structure of the to-be produced speech sequence (either a single vowel or a CV/CVCV sequence) and the class of the vowel. This is coherent with previous findings showing that the muscular activity associated with vowel production is strongly influenced by the surrounding consonants in CVC sequences [78]. Thus, further investigations should focus on how the average EMG amplitude is impacted by coarticulation during the production of CVCV sequences.

With regards to inner speech, our results do not support theoretical predictions of the *motor simulation view*, according to which it should be possible to discriminate classes of nonwords produced in inner speech based on EMG signals. Whereas this outcome is consistent with some recent results [32], it also stands in sharp contrast with classical results in the field [30, 31] as well as more recent developments. For instance, [61] developed a wearable device composed of seven surface EMG sensors that can attain a 92% median classification accuracy in discriminating internally vocalised digits. There are a few crucial differences between [61]'s work and ours that stand as good candidates to explain the discrepancies between our results. First, the strategy adopted to position the sensors was radically different. Following guidelines from the field of psychophysiology, our strategy was to position sensors precisely over the facial muscles of interest, aligned with the direction of the muscle fibers and in theoretically optimal positions to record activity of this muscle while reducing cross-talk. However, precisely because of pervasive cross-talk in facial surface EMG recordings,

this strategy, whereas maximising the probability of recording activity from a given single muscle, was also (as a result) reducing the probability of recording activity from potentially speech-relevant neighbour muscles. Therefore, this strategy might work sub-optimally when the goal of the experiment is to extract the maximum amount of (relevant) EMG information to discriminate inner speech content. However, this problem might be mitigated by using more sensors and a more lenient sensor-positioning approach. Whereas we recorded the EMG amplitude over only two lower facial muscles (OOI and ZIG), [61] analysed EMG data from seven different sensors, whose position and number was defined iteratively in order to maximise the classification accuracy. In other words, the parameters of the experiment were iteratively optimised to maximise a certain outcome (classification accuracy). This strategy is radically different from the classical approach in experimental and cognitive psychology where experimental conditions are defined to test theoretically derived hypotheses. Whereas the first approach is arguably more efficient at solving a particular problem at hand, the second approach might be more efficient in tackling theoretical questions. For instance, a recent study reported a greater EMG amplitude of laryngeal and lip muscles during auditory verbal tasks (covert singing) than during visual imagery tasks [88]. By coupling EMG recording with demographic and psychological measures, they were able to show that these correlates were related to the level of accuracy in singing, thus shedding light upon the nature and functions of peripheral muscular activity during covert singing. However, adding more sensors (e.g., on the risorius), or better optimising sensor placement, could improve the sensitivity of the present approach.

Putting aside considerations related to methodological aspects of the present study, these results do not corroborate the *motor simulation view* of inner speech production. Instead, it seems to support the *abstraction view*, which postulates that inner speech results from the activation of abstract linguistic representations and does not engage the articulatory apparatus. However, individual differences in discriminability highlight that the abstractness of inner speech might be flexible, as suggested by [22]. Indeed, although for most participants it was not possible to decode the phonetic content of inner speech, rounded and spread nonwords were in fact distinguishable based on OOI and ZYG information only (by visual inspection of the 2D plot), for two of them (S_15 and S_17, cf. S1 Text). This suggests either that the extent to which inner speech production recruits the speech motor system might vary between individuals or that it might vary within individual depending on the properties of the ongoing task (these two suggestions are not mutually exclusive). For instance, we know from early research on the EMG correlates of inner speech that the average amplitude of these correlates tend to be higher when the task is more difficult [10]. As such, the extent to which inner speech production recruits the speech motor system could be moderated by manipulating the difficulty of the ongoing task. In addition, the electromyographic activity recorded during motor imagery could be modulated by the perspective taken in motor imagery. A distinction is made between first-person perspective or *internal imagery* (i.e., imagining an action as we would execute it) and third-person perspective or *external imagery* (i.e., imagining an action as an observer of this action), that may involve different neural processes [89]. It has been shown that a first-person perspective may result in greater EMG activity than motor imagery in a third-person perspective [90, 91]. Therefore, we hypothesise that the involvement of the speech motor system during inner speech production may be modulated by the specific instructions given to the participants. For instance, by instructing participants to focus on *inner speaking* (imagining speaking), instead of *inner hearing* (imagining hearing), and by asking them to focus on the kinaesthetic feelings related to speech acts (rather than on auditory percepts), we could expect to find a higher average EMG amplitude recorded over the speech muscles. In addition, by specifically asking the participants to mentally articulate the nonwords, as if they were

dictating them to someone, rather than just read and visually scan them, we may expect stronger articulatory involvement.

Of course, the current study and the above discussion should be interpreted with a few words of caution in mind. For each class of nonwords, we collected around 6 x 10 = 60 observations by condition and by participant. For 25 participants and two classes of nonwords, this results in 25 (participants) x 120 (individual trials) x 3 (conditions) = 9000 observations. However, after rejecting trials with movement artefacts, we had 7285 observations in total. Although the number of observations reported in the present study is reasonable, the sensitivity of the experiment could be improved by increasing the number of observations and/or by reducing two important sources of variation. More precisely, one could reduce the variance related to the item (the specific nonword being uttered) by selecting nonwords that are more similar to each other in the way they are uttered, by selecting less items or simpler items. Similarly, particular attention should be devoted to reducing inter-participant variability, which could be done by using more guided and specific instructions, as well as a longer training phase to familiarise the participant with the task.

In summary, we have demonstrated that whereas surface electromyography may lead to reasonable accuracy in discriminating classes of nonwords during overt speech production (using signals recorded over only two speech-related muscles), it did not permit to discriminate these two classes during inner speech production across all participants (only for two participants). These results, in comparison with results obtained by other teams [61], highlight that depending on the aim of the research, different strategies might be more or less successfully pursued. More precisely, if the goal is to attain high classification accuracy (problem-solving approach), then the parameters of the experiment (e.g., number of repetitions, number of sensors, position of the sensors, parameters of the signal processing workflow) should be optimised based on the desired outcome (i.e., classification accuracy). However, the classical laboratory strategy used in experimental and cognitive psychology, aiming to compare specific conditions (or muscles) to each other in a controlled environment, is deemed to be more appropriate when the aim of the research is to sharpen our understanding of the psychological phenomenon under study.

## Supporting information

**S1 Text.**
(TXT)

## Author Contributions

**Conceptualization:** Ladislas Nalborczyk, Romain Grandchamp, Ernst H. W. Koster, Marcela Perrone-Bertolotti, Hélène Lœvenbruck.

**Data curation:** Ladislas Nalborczyk, Romain Grandchamp, Hélène Lœvenbruck.

**Formal analysis:** Ladislas Nalborczyk, Romain Grandchamp, Hélène Lœvenbruck.

**Funding acquisition:** Ladislas Nalborczyk, Hélène Lœvenbruck.

**Investigation:** Ladislas Nalborczyk, Romain Grandchamp, Ernst H. W. Koster, Marcela Perrone-Bertolotti, Hélène Lœvenbruck.

**Methodology:** Ladislas Nalborczyk, Romain Grandchamp, Ernst H. W. Koster, Marcela Perrone-Bertolotti, Hélène Lœvenbruck.

**Project administration:** Ladislas Nalborczyk, Romain Grandchamp, Ernst H. W. Koster, Hélène Lœvenbruck.

**Resources:** Ladislas Nalborczyk, Romain Grandchamp, Ernst H. W. Koster, Marcela Perrone-Bertolotti, Hélène Lœvenbruck.

**Software:** Ladislas Nalborczyk, Romain Grandchamp.

**Supervision:** Romain Grandchamp, Ernst H. W. Koster, Marcela Perrone-Bertolotti, Hélène Lœvenbruck.

**Validation:** Ladislas Nalborczyk, Ernst H. W. Koster, Marcela Perrone-Bertolotti, Hélène Lœvenbruck.

**Visualization:** Ladislas Nalborczyk.

**Writing – original draft:** Ladislas Nalborczyk.

**Writing – review & editing:** Ladislas Nalborczyk, Romain Grandchamp, Ernst H. W. Koster, Marcela Perrone-Bertolotti, Hélène Lœvenbruck.

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
