## [Decision Letter · Decision Letter 0]

22 Jan 2020

PONE-D-19-27197

Can we decode phonetic features in inner speech using surface electromyography?

PLOS ONE

Dear Mr. Nalborczyk,

Thank you for submitting your manuscript to PLOS ONE. After careful consideration, we feel that it has merit but does not fully meet PLOS ONE’s publication criteria as it currently stands. Therefore, we invite you to submit a revised version of the manuscript that addresses the points raised during the review process.

I have received two reviews from experts in the field and I have also read the paper. Both reviewers and I think the paper is interesting and deserves consideration. The reviewers give helpful suggestions that I would ask you to implement with care. I only add one indication: I agree with reviewer 2 that Vygotskian theory should be cited and discussed in the paper, but I would suggest to include it in the Introduction without re-writing it.

We would appreciate receiving your revised manuscript by Mar 07 2020 11:59PM. To enhance the reproducibility of your results, we recommend that if applicable you deposit your laboratory protocols in protocols.io, where a protocol can be assigned its own identifier (DOI) such that it can be cited independently in the future. For instructions see: http://journals.plos.org/plosone/s/submission-guidelines#loc-laboratory-protocols

We look forward to receiving your revised manuscript.

Kind regards,

Simone Sulpizio

Academic Editor

PLOS ONE

Journal Requirements:

2. We note you have included a table to which you do not refer in the text of your manuscript. Please ensure that you refer to Table 5 in your text; if accepted, production will need this reference to link the reader to the Table.

Reviewers' comments:

Reviewer's Responses to Questions

**Comments to the Author**

1. Is the manuscript technically sound, and do the data support the conclusions?

Reviewer #1: Yes

Reviewer #2: Yes

2. Has the statistical analysis been performed appropriately and rigorously? 

Reviewer #1: Yes

Reviewer #2: Yes

3. Have the authors made all data underlying the findings in their manuscript fully available?

Reviewer #1: Yes

Reviewer #2: Yes

4. Is the manuscript presented in an intelligible fashion and written in standard English?

Reviewer #1: Yes

Reviewer #2: Yes

5. Review Comments to the Author

Reviewer #1: Nalborczyk and colleagues present a study in which they tested the ‘motor simulation’ model of inner speech by measuring EMG to overt speech, inner speech and listening in the OOI and ZYG muscles (and control muscles) in healthy adult volunteers. They did not observe the expected difference in EMG activity between the OOI and ZYG muscles to overt speech, which somewhat complicated the comparison with the inner speech condition. They did find that it was possible to discriminate between the phonetic content of inner speech in some individuals, but this did not maintain at the level of the group.

Overall, I think this this paper represents a nice piece of scholarship. The literature review was comprehensive, well written and argued, and the hypotheses were clear and well justified. The statistics and analyses were sophisticated and thoroughly and carefully presented. I particularly liked how the (key) analyses were pre-registered – IMO this is a good habit and one which should be encouraged in the field. The interpretability of the results was limited somewhat by the study’s failure to replicate the key finding of differential EMG activity in the OOI and ZYG muscles during the (overt) production of rounded vs. spread nonwords. But that’s the way science goes, and it would be good to have these data available in the literature, and I believe that the publication of null results should be encouraged in order to provide a more accurate reflection of the true state of the field.

I have a few minor comments that the authors could consider, primarily regarding the experimental task itself:

- Probably my main point is that I do not understand is the ‘artefact removal procedure’ described on p.8. The authors state: “To remove these signals, we first divided the EMG signals into periods of 1 second” – were these epochs centred around something (e.g., fixation dot onset), or was the start/end location arbitrary? How did the authors distinguish between ‘unwanted’ EMG activity (caused by swallowing, etc.) from ‘of-interest’ EMG activity caused by movement of the articulators? It is possible that the artefact removal procedure removed some of the signal of interest? Could this not be problematic, particularly if the authors only applied the artefact removal procedure to the listening and inner speech conditions? (as seems to be implied on p.8, but I could be wrong here – please clarify).

- P.7. To clarify: am I right in understanding that the EMG activity in the 1 second following the presentation of the fixation dot was used as the dependent variable? If so, were participants instructed to produce the inner/overt speech as soon as possible after the fixation dot appeared? What was the delay (on average) between the dot and speech production in the overt speech condition? Do the authors assume that this delay was consistent between the inner and overt speech conditions? If so, can they justify this assumption?

- It appears as though the ‘baseline’ period (i.e., against which the standardized scores were calculated) was calculated during the relaxation task that occurred prior to the talk itself, is that correct? I don’t understand the benefit of doing this as opposed to, say, calculating the baseline from the 1 second prior to the presentation of the fixation dot – can the authors clarify?

- Figure 1: I think it would be appropriate to mention in the caption that the scales between the figure panels differ markedly – these differences are understandable (i.e., EMG activity in the overt speech condition will obviously be higher than the inner speech / listening conditions), but still appropriate to mention.

Reviewer #2: The present article addresses an important but overlooked question about the external behavioural traces of inner speech. Its reliance on preregistered methods is really impressive, particularly in light of the complexity of possible analyses and the risk of data ‘fishing’. We have several main concerns. One is the theoretical framework in which the study has been conceived, which over-emphasises a problematic distinction between simulation and abstraction views, and pays insufficient attention to the key work in this area.

Abstract. Second sentence should be revised; the authors do not need to describe the steps in detail. The last sentence should also be removed as this information should be submitted separately from the abstract (and also should be included in the Data Availability section at the end of the manuscript).

2. A strong claim is made about the automatic elicitation of inner speech in reading. The picture is not quite so clear-cut; Russell Hurlburt for example has recently published a study showing very little inner speech during reading.

9. The discussion of the featural properties of inner speech is a little thin and could be enhanced.

13. The term ‘inner voice’ is problematic – does it mean ‘inner speech’? In our view they have different connotations (see Fernyhough’s recent book on inner speech, The Voices Within, for a discussion).

18 – 28. This section is incomplete and does not include the essential Vygotskian theory that inner speech is internalised external speech. This is a serious omission. See Alderson-Day and Fernyhough (2015) for a full description of this theory. This is important not just because it is the most developed theory of inner speech, but because it is also highly relevant to the question of whether any behavioural traces of inner speech will be observed (see the discussion in Jones & Fernyhough, 2007, cited here as ref 4, and also extensive discussion in Fernyhough’s book). The fact that Vygotsky’s name is not mentioned in this manuscript is surprising. The introduction will need to be rewritten to represent these important views, which are significant for all of the main hypotheses in this study and for the interpretation of the findings.

Furthermore, since Vygotsky’s is a developmental theory, it is very relevant to the question of different methods and findings for EMG studies of inner speech in children and adults (line 547). Giving some attention to Vygotsky’s theory would also make a lot of sense of the authors’ surprising findings on EMG activity during inner speech. See the discussions cited above about whether, on an internalisation view, you would expect any motoric trace of inner speech, if the latter is fully internalised and transformed outer speech – which also brings in the important issue of semantic and syntactic condensation.

188. The authors describe the sampling procedure in detail, but the Participants section would benefit from including more demographic information on the sample.

214. Studies looking at EMG correlates of lip muscle activity often investigate the orbicularis oris superior muscle alongside the orbicularis oris inferior. The authors should discuss in more detail their justification for studying the OOI, but not OOS.

243. ‘during rumination’ – we think this is a mistake, and an alternative term should be used, e.g., covert speech condition.

245. Please give more detail here on how the stimuli were selected. Were the stimuli piloted before being selected?

253. More detail is needed on the instructions given in the three conditions. Hurlburt et al. (PLOS, 2016) have recently shown a difference in brain activations between elicited and spontaneous inner speech, questioning whether inner speech elicited by the method apparently used here can be taken as a reliable proxy for genuine, spontaneous inner speech. This needs some acknowledgement and discussion, especially as it addresses many of the issues raised in the Jones & Fernyhough (2007) paper.

More explanation is needed on the instruction given to participants during training, e.g. how the visual cues were explained to indicate the start of the task.

349. Figures 6 & 7 could be moved to Supplementary Information.

583-628. The paragraph on stimuli selection should be revised. The detailed information should be moved to the Method section, where it would have been useful to have more information on how the stimuli were selected. More general reflection on this process should be included in the Discussion instead.

6. PLOS authors have the option to publish the peer review history of their article (what does this mean?). If published, this will include your full peer review and any attached files.

Reviewer #1: No

Reviewer #2: No

---

## [Author Response · Author response to Decision Letter 0]

10 Mar 2020

Response to editorial revision requests

We are very grateful to the editor and reviewers for their suggestions and comments which substantially contributed to the improvement of our paper. We provide a point-by-point response in the following and have included appropriate changes in the revised version of our manuscript.

Reviewer #1

Nalborczyk and colleagues present a study in which they tested the ‘motor simulation’ model of inner speech by measuring EMG to overt speech, inner speech and listening in the OOI and ZYG muscles (and control muscles) in healthy adult volunteers. They did not observe the expected difference in EMG activity between the OOI and ZYG muscles to overt speech, which somewhat complicated the comparison with the inner speech condition. They did find that it was possible to discriminate between the phonetic content of inner speech in some individuals, but this did not maintain at the level of the group.

Overall, I think this this paper represents a nice piece of scholarship. The literature review was comprehensive, well written and argued, and the hypotheses were clear and well justified. The statistics and analyses were sophisticated and thoroughly and carefully presented. I particularly liked how the (key) analyses were pre-registered – IMO this is a good habit and one which should be encouraged in the field. The interpretability of the results was limited somewhat by the study’s failure to replicate the key finding of differential EMG activity in the OOI and ZYG muscles during the (overt) production of rounded vs. spread nonwords. But that’s the way science goes, and it would be good to have these data available in the literature, and I believe that the publication of null results should be encouraged in order to provide a more accurate reflection of the true state of the field.

I have a few minor comments that the authors could consider, primarily regarding the experimental task itself:

- Probably my main point is that I do not understand is the ‘artefact removal procedure’ described on p.8. The authors state: “To remove these signals, we first divided the EMG signals into periods of 1 second” – were these epochs centred around something (e.g., fixation dot onset), or was the start/end location arbitrary? How did the authors distinguish between ‘unwanted’ EMG activity (caused by swallowing, etc.) from ‘of- interest’ EMG activity caused by movement of the articulators? It is possible that the artefact removal procedure removed some of the signal of interest? Could this not be problematic, particularly if the authors only applied the artefact removal procedure to the listening and inner speech conditions? (as seems to be implied on p.8, but I could be wrong here – please clarify).

We thank the reviewer for their comment. These elements are indeed crucial in interpreting our data. The 1-second epochs were not chosen arbitrarily but correspond to the periods of interest, that is, the periods during which participants were either 1) doing nothing (that is the baseline period, composed of 60 periods of 1-second), 2) producing overt speech (the “speech” condition, composed of 6 repetitions of each nonword, that is overall 6*20 periods of 1 second per participant), 3) producing inner speech (the “inner speech” condition, composed of 6 repetitions of each nonword, that is overall 6*20 periods of 1 second per participant), or 4) listening to overt speech (the “listening ” condition, composed of 6 repetitions of each nonword, that is overall 6*20 periods of 1 second per participant).

Of course, the artefact rejection procedure could not be applied similarly in the overt speech condition, since it is impossible to visually distinguish speech-related activation from non- speech related activities such as swallowing. But, in addition to the artefact rejection procedure applied to the inner speech and listening conditions, further checks were carried out. In all conditions, since we had recorded the audio signal, any time a non-speech noise (such as coughing or yawning) was present, the EMG signal for that trial was discarded. In the listening and overt speech conditions, if a burst of EMG activity was present after the relevant audio speech signal, then the trial was discarded. In the overt speech condition, if the participant started too early or too late and only part of the nonword was recorded in the audio signal, then the corresponding trial was discarded. Moreover, the fact that the artefact rejection procedure differ in the various conditions is not an issue, since we do not directly compare between conditions. Instead, we compare the EMG correlates of the two classes of nonwords within each condition.

However, we agree it is a possibility that this procedure might remove some of the signal of interest in the inner speech and listening conditions. The reason we still applied it is because the kind of activation we can record in the inner speech condition may be orders of magnitude lower than parasite activations such as activation related to breathing, swallowing, facial expressions, or coughing (amongst others). Therefore, although these “parasite activities” are expected to be distributed uniformly across the two classes of non-words, they add a significant amount of noise to the signal and could diminish the ability to detect inner-speech-related EMG correlates.

To clarify this, we have made the following change in the manuscript (cf. page 9):

Although participants were explicitly asked to remain still during inner speech production or listening, small facial movements (such as swallowing movements) sometimes occurred. Such periods were excluded from the final sample of EMG signals. To remove these signals, we first divided the portions of signals of interest into periods of 1 second. The baseline condition was therefore composed of 60 trials of 1-second. The periods of interest in all the speech conditions consisted of the 1 second interval during which the participants either produced speech or listened to speech. It is possible that the nonword took less than 1 second to be produced, but since there was no way to track when production started and ended in the inner speech condition, the entire 1-second period was kept. Therefore, the overt speech condition was composed of 6 repetitions of each nonword, that is 6*20 trials of 1 second. The “inner speech” and « listening » conditions were similarly composed of 6*20 trials of 1 second. Then, we visually inspected the EMG signals recorded during each trial and listened to the audio signal simultaneously recorded. In all conditions, any time a non-speech noise (such as coughing or yawning) was audible in a trial, the trial was discarded (i.e., we did not include this trial in the final analysis, for any of the recorded muscles). In the listening and overt speech conditions, if a burst of EMG activity was present after the relevant audio speech signal, then the trial was discarded. In the overt speech condition, if the participant started too early or too late and only part of the nonword was recorded in the audio signal, then the corresponding trial was discarded. In the inner speech and listening condition, if a large EMG burst of activity was present, potentially associated with irrelevant non-speech activity, we excluded the trial. The fact that the artefact rejection procedures slightly differ in the various conditions is not an issue, since we do not directly compare between conditions. Instead, we compare the EMG correlates of the two classes of nonwords within each condition.

- P.7. To clarify: am I right in understanding that the EMG activity in the 1 second following the presentation of the fixation dot was used as the dependent variable? If so, were participants instructed to produce the inner/overt speech as soon as possible after the fixation dot appeared? What was the delay (on average) between the dot and speech production in the overt speech condition? Do the authors assume that this delay was consistent between the inner and overt speech conditions? If so, can they justify this assumption?

In both the “speech” and the “inner speech” conditions, the nonword was first visually presented on the screen (for 1 second) and then a fixation dot appeared (for 1 sec), indicating to the participants that they had to produce the nonword that they had just seen (i.e., participants had to produce the nonword in the 1 second time window during which the fixation dot was on the screen). In the “listening” condition however, the order of the fixation dot and the blank screen was reversed. First, the fixation dot appeared for 1 second (so that participants get ready for the

 task, which consists, in this condition, in listening to an overtly uttered nonword), followed by a fixation dot (for 1 second), during the presentation of which the nonword was delivered through the speakers.

Regarding the instructions that were given to the participants, participants were asked to produce the nonword “when the fixation dot appears” in both the “speech” and “inner speech” conditions. Due to inattention or anticipation effects, we might expect that some participants may indeed sometimes “miss” the 1-second time window and produce speech either too late or too soon, respectively. We did not analyse the average “onset time” (i.e., the delay between the apparition of the fixation dot and the beginning of the movement) because these are not available in the inner speech condition (as the EMG signal does not show discernible bursts of activations).

Although we generally assume that the dynamics of overt and covert speech may be roughly similar, it is indeed possible that production in the inner speech condition may start earlier (for instance because there is less articulatory constraints on the production of inner speech, that is, participants do not have to move their articulators to produce sounds). However, given that the purpose of this study is to highlight feature-specific EMG correlates (i.e., it aims at discriminating two classes of non-words), and because we do not have any reason to think that the “dynamics” of (overt and covert) speech production may be affected by this variable (i.e., the non-word class) in any way that could affect our main measure (i.e., the average EMG amplitude), we do not think this could be an issue in the interpretation of our data.

We have therefore clarified this section in our revised manuscript and we have corrected a mistake in this paragraph (cf. lines 361-366). Moreover, we have added this sentence in the description of the periods of interest:

The periods of interest in all the speech conditions consisted of the 1 second interval during which the participants either produced speech or listened to speech. It is possible that the nonword took less than 1 second to be produced, but since there was no way to track when production started and ended in the inner speech condition, the entire 1-second period was kept.

- It appears as though the ‘baseline’ period (i.e., against which the standardized scores were calculated) was calculated during the relaxation task that occurred prior to the talk itself, is that correct? I don’t understand the benefit of doing this as opposed to, say, calculating the baseline from the 1 second prior to the presentation of the fixation dot – can the authors clarify?

We indeed standardised the EMG amplitude in each experimental condition by subtracting the baseline value to it. The main reason we chose to standardise EMG amplitude by baseline activity (rather than pre-stimulus activity) is because in both the “speech” and “inner speech” conditions, the second preceding the overt/inner production of the nonwords was the second during which the nonwords was displayed on the screen, and during which the participants were reading the nonword. Because (some) previous studies have shown an increase in the EMG activity of the lips during reading (e.g., Faaborg-Andersen et al., 1958; Sokolov, 1972), standardising the EMG amplitude during inner production by the activity during reading may obfuscate the activity that could be observed during the task. In other words, subtracting reading (during which inner speech may be produced) to inner speech may result in null activity. Moreover, several previous works have argued for the use of a relaxation period as a baseline, since mere resting periods may include some inner speech production (see e.g., Jacobson, 1931; Vanderwolf, 1998, for a review).

To make this more explicit, we have modified the text as follows:

Baseline EMG measurements were performed during the last minute of this relaxation session, resulting in 60s of EMG signal at baseline. By using this relaxation period as a baseline, we made sure that participants were all in a comparable relaxed state. In addition, several previous EMG studies have argued for the use of a relaxation period as a baseline, since mere resting periods may include some inner speech production (e.g., Jacobson, 1931; Vanderwolf, 1998, for a review).

 Faaborg-Andersen, Edfeldt, K. Å. W. & Nykøbing, F. (1958). Electromyography of Intrinsic and Extrinsic Laryngeal Muscles During Silent Speech: Correlation with Reading Activity: Preliminary Report, Acta Oto-Laryngologica, 49(1), 478-482.

Jacobson, E. (1931). Electrical measurements of neuromuscular states during mental activities. VII. Imagination, recollection, and abstract thinking involving the speech musculature. American Journal of Physiology, 97, 200-209.

Sokolov, A. (1972). Inner speech and thought. New York: Springer-Verlag.

Vanderwolf, C.H. (1998). Brain, behavior, and mind: what do we know and what can we know?

Neuroscience and Biobehavioral Review, 22,125-142.

- Figure 1: I think it would be appropriate to mention in the caption that the scales between the figure panels differ markedly – these differences are understandable (i.e., EMG activity in the overt speech condition will obviously be higher than the inner speech / listening conditions), but still appropriate to mention.

We agree with the reviewer’s comment and we have added a word of caution about this issue in the caption of Figure 1.

Reviewer #2

The present article addresses an important but overlooked question about the external behavioural traces of inner speech. Its reliance on preregistered methods is really impressive, particularly in light of the complexity of possible analyses and the risk of data ‘fishing’. We have several main concerns. One is the theoretical framework in which the study has been conceived, which over-emphasises a problematic distinction between simulation and abstraction views, and pays insufficient attention to the key work in this area.

Abstract. Second sentence should be revised; the authors do not need to describe the steps in detail. The last sentence should also be removed as this information should be submitted separately from the abstract (and also should be included in the Data Availability section at the end of the manuscript).

We thank the reviewer for their suggestion. We have removed the second and last sentences of the abstract.

2. A strong claim is made about the automatic elicitation of inner speech in reading. The picture is not quite so clear-cut; Russell Hurlburt for example has recently published a study showing very little inner speech during reading.

We thank the reviewer for this important comment. We only used “silent reading” as a way to introduce the notion of inner speech in a practical way, as several studies reveal that reading may be accompanied by inner speech (e.g., Yao et al., 2011). However, we fully agree that not everyone produces inner speech during reading (and typically not expert readers, who may well be the typical readers of a scientific paper). We have therefore added a nuance to this introduction, and we inserted the reference suggested by the reviewer (Brouwers et al., 2018), as well as the discussion by Hurlburt (2018). We have also expanded our presentation of the occurrences of inner speech. The manuscript now reads:

As you read these words, you may be experiencing the presence of a familiar speechlike companion. This internal speech production may accompany daily activities such as reading (see [1–4], but see [5,6]), writing ([7]), 4 memorising ([8,9]), future planning [8], problem solving [9,10] or musing (for reviews see [11–14]).

9. The discussion of the featural properties of inner speech is a little thin and could be enhanced.

 We have provided a more thorough description of the featural properties of inner speech and have added references to the literature on this subject. The text is now modified to:

Several studies using experience sampling or questionnaires (e.g.,[14,15]) have shown that by deliberately paying attention to this internal speech, one can examine its phenomenological properties such as identity (whose voice is it?) or other high-level characteristics (e.g., is it gendered?). Moreover, it is often possible to examine lower-level features like the tone of the inner speech, its pitch or its tempo. This set of basic observations leads to some important insights about the nature of inner speech. The simple fact that we can make sensory judgments about our inner speech tautologically reveals that inner speech is accompanied by sensory percepts (e.g., speech sounds, kinaesthetic feelings). Some of these introspective accounts have been examined, tested and complemented using empirical methods from cognitive neuroscience. As summarised in [17], behavioural and neuroimaging data reveal that some variants of inner speech are associated with auditory and/or somatosensory sensations that are reflected by auditory and/or somatosensory cortex activations. Visual representations may also be at play, typically for inner language in the deaf population. Inner verbalising therefore may involve the reception of imaginary multisensory signals. This leads to other fundamental questions: where do these percepts come from? Why do they sound and feel like the ones we experience when we actually (overtly) speak?

13. The term ‘inner voice’ is problematic – does it mean ‘inner speech’? In our view they have different connotations (see Fernyhough’s recent book on inner speech, The Voices Within, for a discussion).

We fully agree that “inner voice” and “inner speech” should not be equated. When we used the term of “inner voice”, we meant the audible voice signal that (may or may not) accompany inner verbal production. When we use the terms of “inner speech”, we mean the activity of talking silently to oneself (where “silently” is to be understood as from the perspective of an external observer, not from the perspective of the inner speaker, for which inner speech may (or may not) be accompanied by sounds). To avoid confusion, we have modified the text and only used the term voice to refer to acoustic voice quality.

18 – 28. This section is incomplete and does not include the essential Vygotskian theory that inner speech is internalised external speech. This is a serious omission. See Alderson-Day and Fernyhough (2015) for a full description of this theory. This is important not just because it is the most developed theory of inner speech, but because it is also highly relevant to the question of whether any behavioural traces of inner speech will be observed (see the discussion in Jones & Fernyhough, 2007, cited here as ref 4, and also extensive discussion in Fernyhough’s book). The fact that Vygotsky’s name is not mentioned in this manuscript is surprising. The introduction will need to be rewritten to represent these important views, which are significant for all of the main hypotheses in this study and for the interpretation of the findings.

Furthermore, since Vygotsky’s is a developmental theory, it is very relevant to the question of different methods and findings for EMG studies of inner speech in children and adults (line 547). Giving some attention to Vygotsky’s theory would also make a lot of sense of the authors’ surprising findings on EMG activity during inner speech. See the discussions cited above about whether, on an internalisation view, you would expect any motoric trace of inner speech, if the latter is fully internalised and transformed outer speech – which also brings in the important issue of semantic and syntactic condensation.

We thank the reviewer for this thoughtful comment. We are well aware of the Vygotskian developmental theory of inner speech and the various extensions that have been proposed in the recent years (see for instance a more exhaustive historical overview in the introduction of Nalborczyk, 2019). However, we do not think the Vygotskian theory of inner speech is directly

 relevant here, as it does not (contrary to the opposition between the motor simulation view and the abstraction view) lead to testable predictions regarding the central question of our experiment, that is, whether the two classes of non-words will be associated with distinct orofacial EMG correlates. In other words, it does not clearly predict whether inner speech will be associated with phonetically-specific “observable” (using EMG) traces in our situation.

We agree, however, that the Vygotskian notion of condensation should be mentioned, as well as the view of Charles Fernyhough and his colleagues that inner speech can vary between two extremes, depending on cognitive demands and emotional conditions. We are aware that condensation is, as claimed in the ConDialInt model by Grandchamp et al. (2019), one of the important dimensions of inner speech. We have therefore substantially modified our text, into the following paragraph in the introduction:

Two main classes of explanatory theories have been offered to answer these questions. A first class of theories, that derives from Vygotky’s views on language and thought, and that we describe as the abstraction view [18] suggest that inner speech is profoundly internalised, abbreviated and condensed in form. Vygotsky suggested that inner speech evolved from so- called egocentric speech (i.e., self-addressed overt speech or private speech), via a gradual process of internalisation during childhood [19]. According to him, the properties of speech are transformed during this internalisation, and inner speech cannot merely be described as a weakened form of overt speech (as claimed for instance by [20]). This has led some scholars to conceive of inner speech as predominantly pertaining to semantics, excluding any phonological, phonetic, articulatory or even auditory properties (e.g., [19–22]). The property of abbreviation and condensation is supported by several psycholinguistic experiments on the qualitative and quantitative differences between overt and covert speech, as concerns rate and error biases (e.g., [20,21,23,24], but see [25]). Such condensation implies that the auditory qualities mentioned above would only rarely be observed during introspection and would merely be the result of learned associations between abstract linguistic representations and auditory percepts. A second class of theories is described under the umbrella term of motor simulation view. These theories suggest that inner speech can be conceived as a kind of action on its own [26,27], produced in the same way as overt speech is, except that the last stage of articulatory execution is only simulated. Most theories under this view share the postulate that the speech motor system is involved (to some extent) during inner speech production and that the auditory and somatosensory consequences of the simulated articulatory movements constitute the inner speech percepts referred to in subjective studies.

As explained in the ConDialInt model [26], these two views can be reconciled if various degrees of unfolding of inner speech are considered. Fully condensed forms of inner speech only involve semantics, and are deprived of the acoustic, phonological and syntactic qualities of overt speech. Expanded forms inner speech, on the other hand, presumably engage prosodic and morpho-syntactic formulation as well as phonological specification, articulatory simulation and the perception of an inner voice. Between the fully condensed abstract forms and the expanded articulation-ready form, it can be assumed that various semi-condensed forms may exist, with morphosyntactic properties and perhaps even phonological features, depending on the stage at which the speech production process is truncated. Such a view was also taken by [29] who has suggested that inner speech varies with cognitive demands and emotional conditions on a continuum between extremely condensed and expanded forms (see also [11,28]). Therefore, the two views of inner speech (abstraction vs. simulation) can be construed as descriptions of two opposite poles on the condensation dimension. On the most expanded side of the continuum, inner speech entails full phonetic specification and articulatory simulation. It might therefore be expected that speech motor activity could be detectable. If the motor simulation view is correct, then motor activity could be recorded during expanded forms of inner speech. If, on the other hand, the abstraction view applies to all forms of inner speech, then no motor activity should be present, even in phonologically-expanded forms.

Nalborczyk, L. (2019). Understanding rumination as a form of inner speech. PhD thesis. Univ. Grenoble Alpes & Ghent University. Retrieved from https://thesiscommons.org/p6dct/

188. The authors describe the sampling procedure in detail, but the Participants section would benefit from including more demographic information on the sample.

 We have added information about the sex and age of our participants (cf. lines 254 -256).

214. Studies looking at EMG correlates of lip muscle activity often investigate the orbicularis oris superior muscle alongside the orbicularis oris inferior. The authors should discuss in more detail their justification for studying the OOI, but not OOS.

We have included more details about this choice in the Methods section (cf. footnote n°1).

Namely, we chose to record only the OOI in this study because we previously observed (e.g., in Nalborczyk et al., 2017; Rapin et al., 2013) that the OOS and OOI were strongly correlated but that the OOI activity was more affected by the experimental manipulation. Given that the information contained in these two muscles was therefore slightly redundant and given material limitations (i.e., we have a limited amount of sensors), we therefore opted for recording additional control muscles (i.e., COR and FRO) instead of the OOS.

Nalborczyk, L., Perrone-Bertolotti, M., Baeyens, C., Grandchamp, R., Polosan, M., Spinelli,

E., ... Lœvenbruck, H. (2017). Orofacial electromyographic correlates of induced verbal rumination. Biological Psychology, 127, 53–63. https://doi.org/10.1016/j.biopsycho.2017.04.013

Rapin, L., Dohen, M., Polosan, M., & Perrier, P. (2013). An EMG Study of the Lip Muscles During Covert Auditory Verbal Hallucinations in Schizophrenia. JSLHR, 56, 1882–1893. https:// doi.org/10.1044/1092- 4388(2013/12- 0210)and

243. ‘during rumination’ – we think this is a mistake, and an alternative term should be used, e.g., covert speech condition.

We thank the reviewer for spotting this mistake, we corrected it.

245. Please give more detail here on how the stimuli were selected. Were the stimuli piloted before being selected?

Stimuli were mostly selected based on theoretical constraints, with the aim of maximising the differences between the two classes of non-words (+ informal piloting on master students). We clarified this in the manuscript (in the “Linguistic material” section). Please note that these theoretical constraints translate well in practice as the two classes of non-words appear to form two clearly distinct clusters in the OOI-ZYG space in the overt speech condition (cf. leftmost panel of Figure 6 in the updated manuscript).

253. More detail is needed on the instructions given in the three conditions. Hurlburt et al. (PLOS, 2016) have recently shown a difference in brain activations between elicited and spontaneous inner speech, questioning whether inner speech elicited by the method apparently used here can be taken as a reliable proxy for genuine, spontaneous inner speech. This needs some acknowledgement and discussion, especially as it addresses many of the issues raised in the Jones & Fernyhough (2007) paper.

More explanation is needed on the instruction given to participants during training, e.g. how the visual cues were explained to indicate the start of the task.

We have clarified the instructions given to the participants in this section (cf. pages 8-9). As made clearer in the introduction (see above), this study only pertains to elicited, wilful inner speech.

349. Figures 6 & 7 could be moved to Supplementary Information.

 We agree that Figure 7 does not contain information that could not be understood in text format and we therefore removed it. However, we do see value in having Figure 6 in the main manuscript, as it allows the reader to immediately see how the EMG amplitude can be clustered in the 2D space formed by the OOI and the ZYG muscles.

583-628. The paragraph on stimuli selection should be revised. The detailed information should be moved to the Method section, where it would have been useful to have more information on how the stimuli were selected. More general reflection on this process should be included in the Discussion instead.

Details concerning the selection of stimuli (of our study) are already provided in the “Linguistic materials” section of our manuscript. It is not clear from the reviewer’s comment what supplementary information should be added to this section.

Please note that the details provided in the paragraph commented on by the reviewer (i.e., lines 583-628 of the previous submission) concern additional analyses performed on the data from Eskes et al. (2017). As this analysis is not central to the purpose of the current study and rather subserves a general reflection (cf. discussion in the subsequent paragraph), we feel this belongs to the discussion section.

---

## [Decision Letter · Decision Letter 1]

4 May 2020

Can we decode phonetic features in inner speech using surface electromyography?

PONE-D-19-27197R1

Dear Dr. Nalborczyk,

We are pleased to inform you that your manuscript has been judged scientifically suitable for publication and will be formally accepted for publication once it complies with all outstanding technical requirements.

With kind regards,

Simone Sulpizio

Academic Editor

PLOS ONE

Additional Editor Comments (optional):

Reviewers' comments:

Reviewer's Responses to Questions

**Comments to the Author**

1. If the authors have adequately addressed your comments raised in a previous round of review and you feel that this manuscript is now acceptable for publication, you may indicate that here to bypass the “Comments to the Author” section, enter your conflict of interest statement in the “Confidential to Editor” section, and submit your "Accept" recommendation.

Reviewer #1: All comments have been addressed

Reviewer #2: All comments have been addressed

2. Is the manuscript technically sound, and do the data support the conclusions?

Reviewer #1: (No Response)

Reviewer #2: Yes

3. Has the statistical analysis been performed appropriately and rigorously? 

Reviewer #1: (No Response)

Reviewer #2: Yes

4. Have the authors made all data underlying the findings in their manuscript fully available?

Reviewer #1: (No Response)

Reviewer #2: Yes

5. Is the manuscript presented in an intelligible fashion and written in standard English?

Reviewer #1: (No Response)

Reviewer #2: Yes

6. Review Comments to the Author

Reviewer #1: (No Response)

Reviewer #2: The authors have thoroughly addressed the concerns raised in the original version of the manuscript.

7. PLOS authors have the option to publish the peer review history of their article (what does this mean?). If published, this will include your full peer review and any attached files.

Reviewer #1: No

Reviewer #2: No

---

## [Editor Report · Acceptance letter]

12 May 2020

PONE-D-19-27197R1 

Can we decode phonetic features in inner speech using surface electromyography? 

Dear Dr. Nalborczyk:

I am pleased to inform you that your manuscript has been deemed suitable for publication in PLOS ONE. Congratulations! Your manuscript is now with our production department. 

With kind regards,

on behalf of

Dr. Simone Sulpizio 

Academic Editor

PLOS ONE